Brief Communication

# Improved reconstruction of single-cell developmental potential with CytoTRACE 2

**Minji Kang** [1,2,3,14], **Gunsagar S. Gulati** [4,14], **Erin L. Brown** [1,2,14], **Zhen Qi** [1,14], **Susanna Avagyan**[2], **Jose Juan Almagro Armenteros**[1,5], **Rachel Gleyzer**[1,2], **Wubing Zhang**[1,2], **Chloé B. Steen**[6,7,8], **Jeremy Philip D'Silva** [1,2], **Janella Schwab** [1,2,9], **Michael F. Clarke** [1,10], **Aadel A. Chaudhuri** [11,12] & **Aaron M. Newman** [1,2,10,13] ✉

While single-cell RNA sequencing has advanced our understanding of cell fate, identifying molecular hallmarks of potency—a cell's ability to differentiate into other cell types—remains a challenge. Here we introduce CytoTRACE 2, an interpretable deep learning framework for predicting absolute developmental potential from single-cell RNA sequencing data. Across diverse platforms and tissues, CytoTRACE 2 outperformed previous methods in predicting developmental hierarchies, enabling detailed mapping of single-cell differentiation landscapes and expanding insights into cell potency.

All cells, from the fertilized egg to its mature progeny, are hierarchically organized in multicellular life. Each cell has distinct potency, or ability to differentiate into specialized cell types, ranging from totipotent (capable of generating an entire organism) and pluripotent (capable of generating all adult cells) to multipotent, oligopotent, unipotent and differentiated cells, each with increasingly restricted developmental potential[1] (Fig. 1a). While lineage tracing, functional transplantation assays and single-cell genomics have expanded our understanding of cell potency[2], there remains a need for interpretable methods that can learn developmental programs, predict potency states and generate insights applicable to regenerative and cancer biology.

We previously introduced CytoTRACE 1 (ref. 3), a computational method for predicting cellular maturity from single-cell RNA sequencing (scRNA-seq) data, based on the number of genes expressed per cell. However, like other trajectory inference methods[4–8], CytoTRACE 1 provides predictions that are dataset-specific, making it difficult to unify results across datasets and contextualize them within the broader framework of cellular potency.

To overcome these challenges, we developed CytoTRACE 2, an interpretable deep learning framework for determining single-cell potency categories and absolute developmental potential from scRNA-seq data. Unlike most deep learning methods[9], CytoTRACE 2 learns multivariate gene expression programs that are readily interpretable and enable accurate predictions of developmental potential. Moreover, it suppresses batch and platform-specific variation through multiple mechanisms, including competing representations of gene expression and training set diversity (Methods). Our approach uncovers cross-tissue correlates of cell potency and highlights the value of interpretable deep learning for characterizing single-cell developmental states in health and disease (https://cytotrace2.stanford.edu).

To develop CytoTRACE 2, we curated an extensive atlas of human and mouse scRNA-seq datasets with experimentally validated potency levels, spanning 33 datasets, nine platforms, 406,058 cells and 125 standardized cell phenotypes (Fig. 1b and Supplementary Table 1). Phenotypes were grouped into six broad potency categories—totipotent, pluripotent, multipotent, oligopotent, unipotent and

[1]Institute for Stem Cell Biology and Regenerative Medicine, Stanford University, Stanford, CA, USA. [2]Department of Biomedical Data Science, Stanford University, Stanford, CA, USA. [3]Department of Computer Science, Stanford University, Stanford, CA, USA. [4]Department of Medical Oncology, Dana-Farber Cancer Institute, Boston, MA, USA. [5]Department of Genetics, Stanford University, Stanford, CA, USA. [6]Department of Medical Genetics, Oslo University Hospital and University of Oslo, Oslo, Norway. [7]Institute for Cancer Research, Oslo University Hospital and University of Oslo, Oslo, Norway. [8]Precision Immunotherapy Alliance, University of Oslo, Oslo, Norway. [9]Department of Bioengineering, Stanford University, Stanford, CA, USA. [10]Stanford Cancer Institute, Stanford University, Stanford, CA, USA. [11]Department of Radiation Oncology, Mayo Clinic, Rochester, MN, USA. [12]Mayo Clinic Comprehensive Cancer Center, Rochester, MN, USA. [13]Chan Zuckerberg Biohub – San Francisco, San Francisco, CA, USA. [14]These authors contributed equally: Minji Kang, Gunsagar S. Gulati, Erin L. Brown, Zhen Qi. ✉e-mail: amnewman@stanford.edu

differentiated—and further subdivided into 24 granular levels based on expected developmental order from lineage tracing and functional assays (Fig. 1b and Supplementary Tables 2 and 3). A training set of 93 cell phenotypes from 16 tissues and 13 studies was used to develop the model, with the remaining data reserved for performance evaluation (Fig. 1b and Supplementary Table 1).

CytoTRACE 2 decodes developmental potential using a novel, explainable deep learning architecture called a gene set binary network (GSBN). Inspired by binarized neural networks[10], GSBNs assign binary weights (0 or 1) to genes, identifying highly discriminative gene sets that define each potency category (Fig. 1c and Extended Data Fig. 1a). Multiple gene sets can be learned for each potency group, and the informative genes driving model predictions can be easily extracted—an advantage over conventional deep learning architectures. As such, CytoTRACE 2 provides two key outputs for each single-cell transcriptome: (1) the potency category with maximum likelihood and (2) a continuous 'potency score' generated by integrating GSBN predictions across potency categories and calibrating the range from 1 (totipotent) to 0 (differentiated) (Fig. 1c, Extended Data Fig. 1a and Supplementary Tables 2–4). Based on the assumption that transcriptionally similar cells occupy related differentiation states, CytoTRACE 2 also leverages Markov diffusion combined with a nearest neighbor approach to smooth individual potency scores (Extended Data Fig. 1b,c).

Having compiled a compendium of ground truth datasets, we evaluated the performance of CytoTRACE 2 by assessing both the accuracy of potency predictions and the ordering of known developmental trajectories. We used two definitions of development ordering: 'absolute order', which compares predictions to known potency levels across datasets, and 'relative order', which ranks cells within each dataset from least to most differentiated (Extended Data Fig. 1d and Supplementary Tables 2–4). The agreement between known and predicted developmental orderings was quantified using weighted Kendall correlation to ensure balanced evaluation and minimize bias (Supplementary Table 5).

We started by evaluating model hyperparameters through cross-validation and observed minimal performance variation across a wide range of values (Extended Data Fig. 1e,f and Supplementary Table 6). Based on this, we selected stable hyperparameters and retrained the model. On the training data, we demonstrated that CytoTRACE 2 achieves high accuracy in distinguishing absolute potency for broad potency labels (Fig. 1d).

To validate our approach, we next extended our analysis to unseen data, comprising 14 held-out datasets spanning nine tissue systems, seven platforms and 93,535 evaluable cells. Performance on broad and granular potency labels was consistently high in testing (Fig. 1d,e) and robust to differences in species, tissues, platforms or phenotypes that were absent during training (Extended Data Fig. 2a–c and Supplementary Table 7). To rigorously assess generalizability, we retrained CytoTRACE 2 on different subsets of the potency atlas, including random train–test splits and scenarios where distinct developmental systems, termed 'clades', were held out from training. In all cases, results were well correlated with ground truth (Fig. 1f, Extended Data Fig. 2d,e and Supplementary Tables 8 and 9), implying that potency-related biology is conserved across datasets. We also found that CytoTRACE 2 is resistant to moderate annotation errors and performs reliably under practical data limitations (Extended Data Fig. 3 and Supplementary Note).

A key advantage of CytoTRACE 2 is its ability to predict absolute developmental potential on a continuous scale from 1 (totipotent) to 0 (differentiated), which enables cross-dataset comparisons and avoids imposing a developmental order where none exists. For example, unlike its predecessor, CytoTRACE 2 corroborated a pluripotency program in cranial neural crest cell precursors[11] and correctly distinguished datasets with and without immature cells[12,13] (Fig. 1g and Extended Data Fig. 4). It also outperformed other methods[3,14–20] in ordering mouse single-cell transcriptomes from six datasets[2,21–25] across 62 developmental time points (Extended Data Fig. 5a–c) and accurately captured the progressive decline in potency across 258 evaluable phenotypes during mouse development (Extended Data Fig. 5d,e)—without requiring data integration or batch correction. CytoTRACE 2 potency predictions also aligned with known leukemic stem cell signatures in acute myeloid leukemia (Extended Data Fig. 6a)[26] and identified known multilineage potential in oligodendroglioma[27], highlighting its applicability to cancer (Extended Data Fig. 6b and Supplementary Table 10).

Next, we benchmarked CytoTRACE 2 against multiple strategies for cell potency classification and developmental hierarchy inference (Supplementary Table 11). CytoTRACE 2 outperformed eight state-of-the-art machine learning methods[28–32] for cell potency classification in 33 datasets, achieving a higher median multiclass F1 score and lower mean absolute error (Extended Data Fig. 7). Moreover, it surpassed eight developmental hierarchy inference methods for cross-dataset (absolute) and intra-dataset (relative) performance[3,14–20], demonstrating over 60% higher correlation, on average, for reconstructing relative orderings in 57 developmental systems, including data from Tabula Sapiens[33] (Fig. 1h,i and Supplementary Tables 12 and 13). Similar results were observed when comparing CytoTRACE 2 against nearly 19k annotated gene sets[34–36] (Fig. 1i and Supplementary Table 13) and scVelo[5], a generalized RNA velocity model for predicting future cell states (Extended Data Fig. 8 and Supplementary Table 14).

Previous genomic studies of stemness largely focused on pluripotency, with limited insight into other potency states. Given the inherent interpretability of our GSBN design, we next explored the molecular programs driving potency predictions (Fig. 2a). Across our potency atlas, GSBN modules produced a cohesive gradient of differentiation states (Fig. 2b and Extended Data Fig. 9a,b). The top-ranking genes showed conserved signatures across species, platforms and developmental clades, identifying both positive and negative correlates of cell potency (Fig. 2c and Supplementary Tables 15 and 16).

Given these results, we hypothesized that CytoTRACE 2 might enrich for key potency-specific factors. Indeed, the core transcription

---

**Fig. 1 | Development and benchmarking of CytoTRACE 2. a**, Overview of cell potency across six developmental categories. **b**, Summary of the 33-dataset single-cell potency atlas. **c**, Schematic of the CytoTRACE 2 model. Toti., totipotent; Pluri., pluripotent; Multi., multipotent; Oligo., oligopotent; Uni., unipotent; Diff., differentiated. **d**, CytoTRACE 2 performance across six broad potency categories in training and held-out test sets, with mean potency scores shown for each standardized phenotype–dataset pair (circles). **e**, CytoTRACE 2 performance across 17 evaluable granular potency levels in held-out test data. Points denote mean potency score per phenotype; large circles indicate the median across these points for each granular potency level. Thick black lines (*x* axis) separate broad potency categories. A linear regression line with 95% confidence band is shown. **f**, Same as **e**, but using a leave-clade-out strategy, where each of 19 developmentally distinct clades (**b**) was held out during training. For **d–f**, concordance with ground truth was assessed using weighted Kendall correlation (τ) applied to single cells, with significance assessed by two-sided *z*-test. Box plots show medians, quartiles and 1.5 × interquartile range (IQR). **g**, Uniform Manifold Approximation and Projection (UMAP) of three held-out datasets showing ground truth (top), CytoTRACE 2 (middle) and CytoTRACE 1 (bottom). **h**, Violin plots comparing nine methods for reconstructing 57 developmental systems. *P* values were calculated by two-sided Wilcoxon tests against CytoTRACE 2; **\**P* < 0.01; *\*\*\**P* < 0.0001. **i**, Performance comparison with eight previous methods and 18,706 gene sets in the test set (left) and Tabula Sapiens (right) using weighted τ to assess absolute (six broad potency levels) and relative order (median correlation across individual trajectories). **a** and **c** were created using BioRender.com.

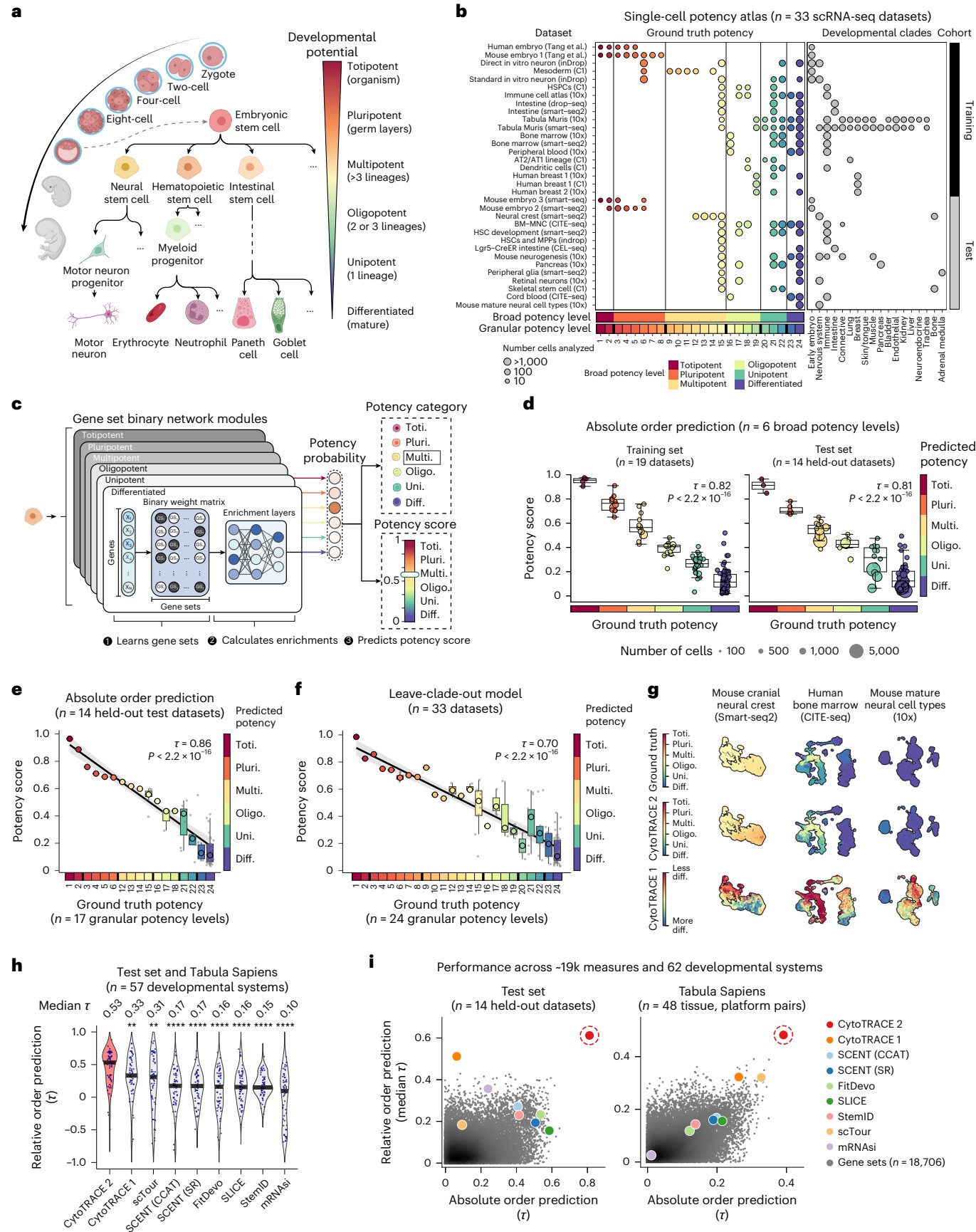

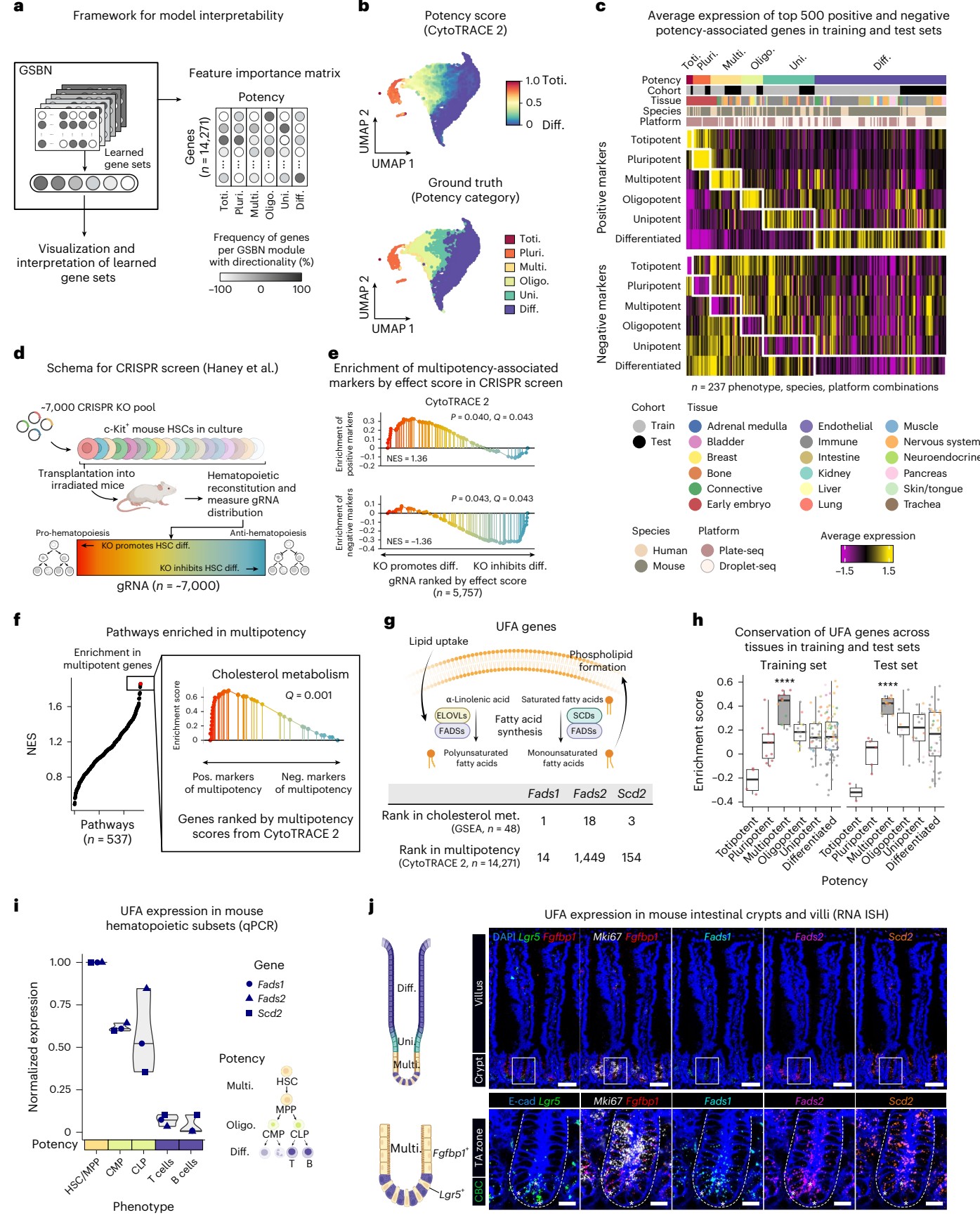

**Fig. 2 | Model interpretability and cross-tissue signatures of cell potency.**
**a**, Schematic for characterizing CytoTRACE 2 gene sets and feature importance.
**b**, UMAP of gene set expression levels in training–test sets, aggregated in a
0.5 × 0.5 grid, colored by CytoTRACE 2 (top) or ground truth potency (bottom).
**c**, Expression of top 500 positive (pos.) and negative (neg.) markers per potency
category, shown across 237 pseudo-bulks aggregated by phenotype, species
and platform from training–test sets. **d**, Overview of a CRISPR knockout (KO)
screen assessing in vivo differentiation effects in hematopoietic stem cells
(HSCs)[38]. **e**, Enrichment of top CytoTRACE 2 multipotency markers among genes
whose knockout promotes or inhibits HSC differentiation (from **d**), using GSEA.
**f**, GSEA of 537 pathways in genes ranked by multipotency scores, highlighting
'cholesterol metabolism'. **g**, Top: overview of UFA pathways, inspired by ref. 42.
Bottom: top UFA biosynthesis genes (*Fads1, Fads2* and *Scd2*) ranked by GSEA and
CytoTRACE 2 multipotency scores). **h**, Single-sample GSEA of UFA genes across
237 pseudo-bulk samples, colored by tissue type as in **c**. ****$P$ < 0.0001 (one-sided
permutation testing). Box plots show medians, quartiles and 1.5 × IQR. **i**, qPCR of
UFA genes in FACS-purified mouse hematopoietic subsets ($n$ = 3), normalized to
HSC/MPP; *Actb* as internal control. MPP, multipotent progenitor; CMP, common
myeloid progenitor; CLP, common lymphoid progenitor. Violin plots show
median and range. **j**, In situ mRNA imaging of mouse jejunum (top) shows spatial
expression of multipotent (*Lgr5* and *Fgfbp1*), proliferation (*Mki67*), and UFA
(*Fads1, Fads2* and *Scd2*) marker genes in crypts and villi. Higher magnification
views (bottom) highlight boxed regions. Cell boundaries were visualized with
E-cadherin immunostaining; asterisks mark representative *Lgr5*⁺ crypt base
columnar (CBC) cells. TA, transit-amplifying. Scale bars, 50 μm (top), 10 μm
(bottom). Images are representative of three mice. Images in **a, d, g, i, j** were
created using BioRender.com. NES, normalized enrichment score.

factors *Pou5f1* and *Nanog*[37] ranked within the top 0.2% of pluripotency
genes (Supplementary Table 15). To further explore this hypothesis,
we analyzed data from a large-scale CRISPR screen, in which ~7,000
genes in multipotent mouse hematopoietic stem cells were individually
knocked out and assessed for developmental consequences in vivo[38]
(Fig. 2d). Among the 5,757 genes overlapping CytoTRACE 2 features,
the top 100 positive multipotency markers were enriched for genes
whose knockout promotes differentiation, whereas the top 100 nega-
tive markers were enriched for genes whose knockout inhibits dif-
ferentiation ($Q$ = 0.04; Fig. 2e and Extended Data Fig. 9c). This trend
was consistent across different numbers of top markers and highly
specific for multipotency, underscoring the fidelity of learned potency
representations (Extended Data Fig. 9d).

To more deeply analyze multipotency in mouse and human tissues
and explore the potential of CytoTRACE 2 for biomarker discovery, we
next applied pathway enrichment analysis to genes ranked by feature
importance. Remarkably, cholesterol metabolism emerged as a leading
multipotency-associated pathway (Fig. 2f, Extended Data Fig. 9e and
Supplementary Table 17). Within this pathway, three genes related to
unsaturated fatty acid (UFA) synthesis (*Fads1, Fads2* and *Scd2*) were
among the top-ranking markers (Fig. 2g). These genes were consistently
enriched in multipotent cells across 125 phenotypes in our potency
atlas (Fig. 2h; train–test area under the curve (AUC) values of 0.87 and
0.92, respectively).

To experimentally confirm these findings, we performed quantita-
tive PCR on mouse hematopoietic cells sorted into multipotent, oligo-
potent, and differentiated subsets (Fig. 2i and Extended Data Fig. 10a,b)
and multiplexed in situ mRNA imaging on mouse intestinal epithelium
co-stained with multipotency markers, *Lgr5*[39] and *Fgfbp1*[40] (Fig. 2j and
Extended Data Fig. 10c–e). In both approaches, *Fads1, Fads2* and *Scd2*
showed reproducible and preferential expression in multipotent cells
(Fig. 2i,j and Extended Data Fig. 10). While fatty acid metabolism has
been linked to stem cell biology[41], no study has specifically attrib-
uted lipid metabolism genes to distinct potency levels. Therefore,
CytoTRACE 2 provides a framework to uncover molecular relationships
and facilitate new hypotheses and discoveries.

In summary, CytoTRACE 2 is an interpretable deep learning
framework that predicts cell potency and continuous differentia-
tion states from scRNA-seq data. Unlike previous methods, it links
stemness and pseudotime to absolute developmental potential, offer-
ing cross-dataset compatibility and transparency into the molecular
profiles driving its predictions. Nonetheless, this study has several limi-
tations. Like all supervised machine learning approaches, CytoTRACE
2 depends on the quality and breadth of its training data, although
robust results were observed across diverse training–test splits, and
moderate labeling variation was well tolerated. Performance may
decline when analyzing cells with very low RNA content or number
of expressed genes (Extended Data Fig. 3). While some phenotypes
were misclassified in held-out data, absolute errors remained low and
outcompeted existing methods. Finally, although the current model is
trained on human and mouse data, ortholog mapping may expand its
applicability to other species. Given its demonstrated advantages, we
anticipate that CytoTRACE 2 will have immediate utility for improving
our understanding of cell potency, with implications for the identifi-
cation of novel biomarkers and therapeutic targets in diseases where
altered developmental hierarchies play a role.

## Online content

Any methods, additional references, Nature Portfolio reporting sum-
maries, source data, extended data, supplementary information,
acknowledgements, peer review information; details of author contri-
butions and competing interests; and statements of data and code avail-
ability are available at https://doi.org/10.1038/s41592-025-02857-2.

## References

1. Zakrzewski, W., Dobrzyński, M., Szymonowicz, M. & Rybak, Z.
   Stem cells: past, present, and future. *Stem Cell Res. Ther.* **10**, 68
   (2019).
2. Qiu, C. et al. A single-cell time-lapse of mouse prenatal
   development from gastrula to birth. *Nature* **626**, 1084–1093
   (2024).
3. Gulati, G. S. et al. Single-cell transcriptional diversity is a hallmark
   of developmental potential. *Science* **367**, 405–411 (2020).
4. La Manno, G. et al. RNA velocity of single cells. *Nature* **560**,
   494–498 (2018).
5. Bergen, V., Lange, M., Peidli, S., Wolf, F. A. & Theis, F. J.
   Generalizing RNA velocity to transient cell states through
   dynamical modeling. *Nat. Biotechnol.* **38**, 1408–1414 (2020).
6. Qiu, X. et al. Reversed graph embedding resolves complex
   single-cell trajectories. *Nat. Methods* **14**, 979–982 (2017).
7. Lange, M. et al. CellRank for directed single-cell fate mapping.
   *Nat. Methods* **19**, 159–170 (2022).
8. Weiler, P., Lange, M., Klein, M., Pe'er, D. & Theis, F. CellRank 2:
   unified fate mapping in multiview single-cell data. *Nat. Methods*
   **21**, 1196–1205 (2024).
9. Rudin, C. Stop explaining black box machine learning models
   for high stakes decisions and use interpretable models instead.
   *Nat. Mach. Intell.* **1**, 206–215 (2019).
10. Hubara, I., Courbariaux, M., Soudry, D., El-Yaniv, R. & Bengio, Y.
    Binarized neural networks. In *Advances in Neural Information
    Processing Systems 29* (eds Lee, D. et al) (Curran Associates,
    2016).
11. Zalc, A. et al. Reactivation of the pluripotency program precedes
    formation of the cranial neural crest. *Science* **371**, eabb4776
    (2021).
12. Stuart, T. et al. Comprehensive integration of single-cell data. *Cell*
    **177**, 1888–1902.e1821 (2019).
13. Zheng, X. et al. Massively parallel in vivo Perturb-seq reveals
    cell-type-specific transcriptional networks in cortical
    development. *Cell* **187**, 3236–3248 e3221 (2024).

14. Teschendorff, A. E., Maity, A. K., Hu, X., Weiyan, C. & Lechner, M. Ultra-fast scalable estimation of single-cell differentiation potency from scRNA-seq data. *Bioinformatics* **37**, 1528–1534 (2020).

15. Teschendorff, A. E. & Enver, T. Single-cell entropy for accurate estimation of differentiation potency from a cell's transcriptome. *Nat. Commun.* **8**, 15599 (2017).

16. Herman, J. S., Sagar & Grün, D. FateID infers cell fate bias in multipotent progenitors from single-cell RNA-seq data. *Nat. Methods* **15**, 379–386 (2018).

17. Guo, M., Bao, E. L., Wagner, M., Whitsett, J. A. & Xu, Y. SLICE: determining cell differentiation and lineage based on single cell entropy. *Nucleic Acids Res.* **45**, e54 (2017).

18. Malta, T. M. et al. Machine learning identifies stemness features associated with oncogenic dedifferentiation. *Cell* **173**, 338–354. e315 (2018).

19. Li, Q. scTour: a deep learning architecture for robust inference and accurate prediction of cellular dynamics. *Genome Biol.* **24**, 149 (2023).

20. Zhang, F. et al. FitDevo: accurate inference of single-cell developmental potential using sample-specific gene weight. *Brief. Bioinform.* **23**, bbac293 (2022).

21. Cheng, S. et al. Single-cell RNA-seq reveals cellular heterogeneity of pluripotency transition and x chromosome dynamics during early mouse development. *Cell Rep.* **26**, 2593–2607.e2593 (2019).

22. Deng, Q., Ramsköld, D., Reinius, B. & Sandberg, R. Single-cell RNA-seq reveals dynamic, random monoallelic gene expression in mammalian cells. *Science* **343**, 193–196 (2014).

23. Mohammed, H. et al. Single-cell landscape of transcriptional heterogeneity and cell fate decisions during mouse early gastrulation. *Cell Rep.* **20**, 1215–1228 (2017).

24. Pijuan-Sala, B. et al. A single-cell molecular map of mouse gastrulation and early organogenesis. *Nature* **566**, 490–495 (2019).

25. Qiu, C. et al. Systematic reconstruction of cellular trajectories across mouse embryogenesis. *Nat. Genet.* **54**, 328–341 (2022).

26. Zeng, A. G. X. et al. A cellular hierachy framework for understanding heterogeneity and predicting drug response in acute myeloid leukemia. *Nat. Med.* **28**, 1212–1223 (2022).

27. Tirosh, I. et al. Single-cell RNA-seq supports a developmental hierarchy in human oligodendroglioma. *Nature* **539**, 309–313 (2016).

28. Abdelaal, T. et al. A comparison of automatic cell identification methods for single-cell RNA sequencing data. *Genome Biol.* **20**, 194 (2019).

29. Cao, X. et al. A systematic evaluation of supervised machine learning algorithms for cell phenotype classification using single-cell RNA sequencing data. *Front. Genet.* **13**, 836798 (2022).

30. Alquicira-Hernandez, J., Sathe, A., Ji, H. P., Nguyen, Q. & Powell, J. E. scPred: accurate supervised method for cell-type classification from single-cell RNA-seq data. *Genome Biol.* **20**, 264 (2019).

31. Tan, Y. & Cahan, P. SingleCellNet: a computational tool to classify single cell RNA-seq data across platforms and across species. *Cell Syst.* **9**, 207–213 e202 (2019).

32. Kiselev, V. Y., Yiu, A. & Hemberg, M. scmap: projection of single-cell RNA-seq data across data sets. *Nat. Methods* **15**, 359–362 (2018).

33. Consortium, T. T. S. et al. The Tabula Sapiens: a multiple-organ, single-cell transcriptomic atlas of humans. *Science* **376**, eabl4896 (2022).

34. Gerstein, M. B. et al. Architecture of the human regulatory network derived from ENCODE data. *Nature* **489**, 91–100 (2012).

35. Lachmann, A. et al. ChEA: transcription factor regulation inferred from integrating genome-wide ChIP-X experiments. *Bioinformatics* **26**, 2438–2444 (2010).

36. Liberzon, A. et al. The Molecular Signatures Database hallmark gene set collection. *Cell Syst.* **1**, 417–425 (2015).

37. Loh, Y. H. et al. The Oct4 and Nanog transcription network regulates pluripotency in mouse embryonic stem cells. *Nat. Genet.* **38**, 431–440 (2006).

38. Haney, M. S. et al. Large-scale in vivo CRISPR screens identify SAGA complex members as a key regulators of HSC lineage commitment and aging. Preprint at *bioRxiv* https://doi.org/10.1101/2022.07.22.501030 (2022).

39. Barker, N. et al. Identification of stem cells in small intestine and colon by marker gene Lgr5. *Nature* **449**, 1003–1007 (2007).

40. Capdevila, C. et al. Time-resolved fate mapping identifies the intestinal upper crypt zone as an origin of Lgr5+ crypt base columnar cells. *Cell* **187**, 3039–3055 e3014 (2024).

41. Kang, J. X., Wan, J. B. & He, C. Concise review: regulation of stem cell proliferation and differentiation by essential fatty acids and their metabolites. *Stem Cells* **32**, 1092–1098 (2014).

42. Jin, H.-R. et al. Lipid metabolic reprogramming in tumor microenvironment: from mechanisms to therapeutics. *J. Hematol. Oncol.* **16**, 103 (2023).

## Methods

### Ethical compliance

All animal procedures were performed in compliance with ethical regulations and conducted according to a protocol approved by the Stanford University Administrative Panel for Laboratory Animal Care committee (protocol no. 10868).

### Single-cell potency atlas

Developmental potency reflects a cell's capacity to differentiate into various cell types, with six widely recognized categories in stem cell biology: totipotency, pluripotency, multipotency, oligopotency, unipotency, and differentiated (Fig. 1a,b and Supplementary Note). These broad classifications are based on decades of research, including lineage tracing, transplantation and colony-formation experiments across multiple tissues and species. Each category represents a progressively restricted ability to generate downstream cell types, from totipotent cells capable of forming all embryonic and extra-embryonic lineages to unipotent cells restricted to producing a single mature cell type; however, as developmental potential exists on a continuum, we also devised a more granular classification system, as described in Supplementary Note and Supplementary Tables 2 and 3.

Of note, classically defined potency levels are not directly annotated in publicly available scRNA-seq datasets. Therefore, to train, validate and benchmark CytoTRACE 2, we downloaded and curated 33 human and mouse scRNA-seq datasets from peer-reviewed studies with experimentally confirmed developmental states and assignable potency levels (Supplementary Table 1). As part of this selection process, we applied the following inclusion and exclusion criteria to enhance experimental rigor:

- Only functionally validated developmental states supported by lineage tracing or transplantation assays were considered for analysis. Datasets with transient cell changes, such as from metabolic activation or suppression, cell cycle transitions or environmental perturbations were excluded, as these do not represent durable developmental processes.
- Datasets with irreconcilable technical batches resulting in major imbalances in the number of cells per phenotype were excluded.
- Single-nucleus RNA sequencing datasets were excluded, as they do not capture cytoplasmic RNA and include immature transcripts.

Among datasets satisfying these conditions, author-supplied cell type annotations were mapped to one of six broad potency categories (totipotent, pluripotent, multipotent, oligopotent, unipotent and differentiated) or not evaluable using established definitions ('Potency annotation scheme', Supplementary Note). These potency categories were further subdivided into 24 granular categories, ranging from 1 (least differentiated) to 24 (most differentiated) (Supplementary Tables 2 and 3). Cellular phenotypes were hierarchically grouped into these categories based on potency, developmental timing and sequence, and self-renewal capacity.

Where possible, we also examined single-cell developmental states in a dataset-specific manner and without regard to potency categories, as previously described[3]. Such 'relative' orderings, most of which were obtained from Gulati et al.[3], ranged from 1 (least differentiated) to $N$ (most differentiated) in a given dataset, and exceeded the number of resolvable potency categories in some datasets (Supplementary Table 4), permitting a more granular assessment

Our comprehensive potency atlas catalogs experimentally confirmed cell states and their corresponding potency levels, providing a structured reference for model training and validation. Supplementary Table 3 includes key details such as the broad and granular potency levels, standardized and original cell phenotypes, species, dataset source, cohort type (for example, training, validation and test), developmental maturity, lineage contributions and supporting evidence. This format allows for consistent annotation and comparison across datasets. For full details of potency annotations and associated rationale, see 'Potency annotation scheme' (Supplementary Note) and Supplementary Tables 2–4.

**Training and test datasets.** Using the abovementioned criteria, we assembled a 33-dataset potency atlas (Fig. 1b), from which we selected a training cohort consisting of seven human and 12 mouse scRNA-seq datasets from 13 studies (Supplementary Table 1). We ensured that all six broad potency categories were represented in both species along with a diverse array of biological (for example, tissue types) and technical characteristics (for example, sequencing platforms). As part of this effort, and to align with precedent in the field, we incorporated all human and mouse scRNA-seq datasets ($n$ = 13) with annotatable potency categories analyzed by Gulati et al.[3]. To broadly cover tissue types, we also included cell phenotypes from the Tabula Muris scRNA-seq atlas[43] for which potency categories could be determined (15 tissue types and 43 phenotypes). The resulting training cohort encompasses 312,523 cells, 16 tissue types, 93 phenotypes and six scRNA-seq platforms (Fig. 1b).

The remaining datasets served as a held-out test cohort, which mirrors the training cohort with respect to species representation in each broad potency category (Supplementary Table 1). Consisting of three human and 11 mouse scRNA-seq datasets from 14 studies, the test cohort spans 93,535 cells, 73 phenotypes, nine tissue types and seven scRNA-seq platforms, including two tissue types and 21 phenotypes that were absent from training (Fig. 1b and Supplementary Tables 1 and 7).

To augment these data, we annotated potency categories in 459,320 evaluable cells from Tabula Sapiens, a multi-tissue scRNA-seq atlas from postmortem human donor biopsies[33] (Supplementary Table 1); however, given the confounding influence of postmortem intervals on human tissue messenger RNA levels[44], we hypothesized that Tabula Sapiens might exhibit reduced data quality. To test this, we calculated the ratio of mitochondrial reads to total reads (MTR) within each single-cell transcriptome as a proxy for overall data quality. Indeed, we calculated a mean MTR across all Tabula Sapiens tissue types, stratified by platform, of 7.4% (median of medians), which is nearly 90% higher than expected for human cell types profiled by scRNA-seq data (median of medians of 3.9%; Table S1 of Osorio and Cai[45]) and 78% higher than other human datasets in the training and test cohorts, both of which include embryonic tissues with high metabolic activity (median of medians of 4.2%). Accordingly, we omitted Tabula Sapiens from the primary test cohort and evaluated it as a secondary benchmark in Fig. 1h,i. Author-supplied phenotypes in Tabula Sapiens with fewer than five cells in a tissue–platform pair were excluded from further analysis.

Collectively, these ground truth datasets with newly annotated potency levels represent a unique community resource for systematic characterization of absolute developmental states and their molecular programs in humans and mice. Depending on platform, all scRNA-seq expression matrices were normalized to transcripts per million (TPM) or counts per million (CPM) as appropriate. Full details of each dataset, including dataset name, accession number, PMID, species, platform, tissue type, number of cells, number of phenotype, and number of potency levels, are available in Supplementary Table 1. These data can also be interactively explored at https://cytotrace2.stanford.edu.

**Additional annotation considerations.** For cells with identical phenotypes but different author-supplied labels, we unified the annotations (Supplementary Table 3). For example, 'HSC-MPPs' from 'HSC development (Smart-seq2)' and 'Hematopoietic stem cell progenitor (HSCP)' from 'HSPCs (C1)' were annotated as 'Hematopoietic stem and early progenitor'. To balance the representation of cells from distinct lineages

within a given broad potency category, we also re-annotated related cell subsets sharing a common parental phenotype. For example, 'CD4+ helper T cells' from 'peripheral blood (10x)' and 'CD8+ memory T cells' from 'BM-MNC (CITE-seq)' were labeled as 'T cell'. This was crucial when training CytoTRACE 2 as the probability of sampling individual cells was weighted based on phenotype. In this way, each major phenotype contributed equally during model training regardless of the number of evaluable cells, mitigating the chance of overweighting and overfitting (see 'Training and hyperparameter tuning' below). The standardized phenotype assignments along with the original annotations are summarized in Supplementary Table 3.

### The CytoTRACE 2 framework

Existing RNA-based surrogates of cellular differentiation status have notable limitations for imputing absolute differentiation states and potency categories from scRNA-seq data. For example, the original CytoTRACE, termed CytoTRACE 1 in this work, employs gene counts as an unbiased strategy for identifying immature cells[3]. Despite the utility of this approach, gene counts are subject to dataset-specific biases, making them suboptimal for potency assessment. Measures based on transcriptional entropy and RNA velocity also suffer from dataset-specific biases, a nonspecific relationship to absolute differentiation status, or the requirement for continuous developmental processes within a narrowly defined time window[4,5,14–16].

Supervised machine learning models offer a potentially robust alternative to the abovementioned strategies when adequate training data are available; however, machine learning methods also face key challenges when applied to scRNA-seq data, including sparsity, high dimensionality and data heterogeneity encompassing both biological and technical variation. While deep learning is a promising subtype of machine learning, often achieving remarkable performance gains over other machine learning methods (especially in the presence of high complexity, noise and uncertainty) most existing architectures lack inherent interpretability, limiting their broad applicability.

To address these challenges, we designed a novel deep learning framework that can handle the complexities of single-cell potency assessment while achieving direct biological interpretability. Unlike recent methods[46,47] that decompose single-cell expression data into a combination of previously known and simultaneously learned new gene programs, our approach, termed a GSBN, is anchored to known phenotypic states but not known gene sets. As such, GSBNs have the flexibility to discover new gene programs for known phenotypic states, such as potency categories, from scRNA-seq data. As part of their design, GSBNs are highly robust and fully interpretable, meaning they can be directly interrogated to extract meaningful markers for each phenotypic class of interest across datasets, platforms and tissues.

**Technical description.** CytoTRACE 2 consists of five high-level components, schematically depicted in Fig. 1c and Extended Data Fig. 1a and described in detail below.

- Preprocessing: ortholog mapping and expression normalization.
- GSBNs: identification of interpretable potency-associated gene sets for each potency category.
- Enrichment assessment: evaluation of gene set activation levels in single cells.
- Integration of scores: integration of gene set activation levels, both within and across gene set binary networks.
- Postprocessing: leveraging transcriptional covariance and uncertainty in model predictions to smooth single-cell potency scores and produce the final output.

**Core model architecture.** Among these five components, GSBNs, enrichment assessment and integration of scores constitute the CytoTRACE 2 core model, a neural network architecture consisting

of a shared input layer; a set of $G$ GSBN modules, where $G$ denotes the number of potency categories; and a shared output layer (Extended Data Fig. 1a). Within the core model, each GSBN module is trained to discriminate a single potency category and contains (1) a binary neural network (BNN) component, which encodes potency-associated gene sets and (2) downstream functions to calculate and integrate gene set enrichment scores (Fig. 1c and Extended Data Fig. 1a). Notably, because weights in BNNs are constrained to binary rather than continuous values, BNNs also allow for more efficient computation and provide an implicit form of model regularization[48].

**Preprocessing.** Let input scRNA-seq dataset $\mathbf{X}$ be an $I \times C$ gene expression matrix over $I$ genes and $C$ cells. The following preprocessing steps prepare the input dataset for training or prediction.

First, gene symbols in $\mathbf{X}$ are mapped and filtered using dictionary $\mathbb{D}$, a collection of gene symbols that harmonizes all HGNC (human) and MGI (mouse) identifiers supported by CytoTRACE 2 ('Dictionary of input genes' below). Following this step, the resulting expression matrix, denoted $\mathbf{X}'$, consists of $n = 14,271$ genes and $C$ cells. As part of this process, any genes in $\mathbf{X}'$ not present in $\mathbf{X}$ through mapping are set to zero. In the second step, $\mathbf{X}'$ is converted into dual representations: for the first, it is normalized to CPM/TPM and $\log_2$-adjusted, yielding an $N \times C$ matrix $\mathbf{L}$; for the second, it is mapped to rank space, yielding an $N \times C$ matrix $\mathbf{R}$, with the genes of each single-cell transcriptome $\mathbf{X}'_c$ assigned relative integer rank such that rank 1 corresponds to the gene with highest expression. While the $\log_2$ CPM/TPM representation maintains detailed transcriptomic information, the alternative encoding provided by rank space helps circumvent batch effects, mitigate the influence of extreme values and outliers, and reduce the risk of model overfitting. In tandem, these two representations provide an inherent regularization to model inputs. $\mathbf{R}$ and $\mathbf{L}$ are subsequently passed to the CytoTRACE 2 core model where they jointly constitute the model input layer.

**Gene set binary networks.** Inputs $\mathbf{R}$ and $\mathbf{L}$ are passed to each of $G$ GSBN modules within the CytoTRACE 2 core model. These modules begin by thresholding $\mathbf{R}$ (Extended Data Fig. 1a) to learnable maximum rank $\tau \in \mathbb{N}$, yielding $N \times C$ matrix $\mathbf{T}$:

$$\mathbf{T}_{i,k} = \min(\mathbf{R}_{i,k}, \tau)$$

This rank trimming (see also 'Model initialization and updates') enables calculation of the rank-based enrichment score, described in 'Enrichment assessment' below. Input $\mathbf{L}$ remains the same.

Next, within each GSBN module, $M$ gene sets are learned in binary $N \times M$ matrix $\mathbf{W}^{\mathrm{B}}$, where $M \in \mathbb{N}$ is prespecified and all entries $\mathbf{W}^{\mathrm{B}}_{i,j} \in \{0,1\}$. $\mathbf{W}^{\mathrm{B}}$ constitutes the gene set selection layer of the CytoTRACE 2 core model; it has a continuous equivalent $\mathbf{W}$ used for model initialization and backpropagation (see also 'Training and hyperparameter tuning'). At each forward iteration for model training, $\mathbf{W}$ undergoes binarization:

$$\mathbf{W}^{\mathrm{B}} = \text{binarize}(\mathbf{W}, 0)$$

where binarize denotes the following utility function:

$$\text{binarize}(\mathbf{M}, a)_{i,j} = \begin{cases} 1, & \mathbf{M}_{i,j} > a \\ 0, & \mathbf{M}_{i,j} \le a \end{cases}$$

**Enrichment assessment.** To quantify the enrichment of each gene set in the module (each column of $\mathbf{W}^{\mathrm{B}}$), CytoTRACE 2 leverages two complementary measures: rank-based enrichment score ($\text{Score}_U$) and expression-based enrichment score ($\text{Score}_A$). $\text{Score}_U$ aggregates overall expression activity of a given gene set $j$ in rank space whereas $\text{Score}_A$ compares the average expression of genes in $j$ versus background levels. By integrating both scores, each providing a different axis of

information, CytoTRACE 2 can learn more complex expression patterns while also achieving additional regularization through enrichment score competition. The two scores are defined as follows.

$Score_U$ calculates the commonly used nonparametric UCell score[49] for each gene set, or column of $\mathbf{W}^B$. For each cell $1 \le k \le C$ and module gene set $1 \le j \le M$,

$$Score_U(\mathbf{T}, \mathbf{W}^B)_{k,j} = 1 + \left[ \frac{\bar{S}_j(\bar{S}_j + 1) - 2\sum_{i=1}^{N} \mathbf{T}_{i,k}\mathbf{W}^B_{i,j}}{2\tau\bar{S}_j} \right],$$

where $\bar{S}$ denotes the vector of length $M$ containing the number of genes per gene set assigned nonzero weight in the binary weighting matrix:

$$\bar{S}_j = \sum_{i=1}^{N} \mathbf{W}^B_{i,j} \quad 1 \le j \le M$$

$Score_A$ implements a scoring system based on Seurat's AddModuleScore (AMS), computing the average expression of genes within a gene set subtracted by the aggregated expression of control, or background, feature sets[50]. To select background features, AMS groups genes into $n_{bins}$ bins according to their average expression within a dataset. Then, for each gene, a 'background' set of $n_{sample}$ genes from the same average expression bin is sampled, ensuring that each gene is compared to other genes with similar average expression. Here, for computational efficiency and to avoid introducing a dependency on dataset composition, we use our entire curated training cohort (see 'Single-cell potency atlas') as the 'dataset' in which to rank genes by average expression. We then compute a constant set of background genes to use for each gene. We encode the mapping of genes to their background genes in the binary $N \times N$ matrix $\mathbf{G}$, where each row represents a gene as used in a gene set, and the $j$th entry of row $i$ is 1 if gene $j$ is used as background for gene $i$, and 0 otherwise.

In detail, we construct $\mathbf{G}$ as follows. First, we compute the average $\log_2$ CPM/TPM expression per gene across all cells from the training cohort. We then rank the results and uniformly partition genes into $n_{bins} = 24$ bins of size $s_{bin}$ according to rank, following the Seurat default[50]. Next, for each gene (each row of $\mathbf{G}$), we randomly select without replacement a set of background genes, where the number of background genes follows a Gaussian distribution with mean $\mu = n_{sample}$ and variance

$$\sigma^2 = n_{sample} \left( \frac{s_{bin} - n_{sample}}{s_{bin}} \right)$$

where $n_{sample} = 100$. This approach provides an additional regularizing effect compared to constant selection of a uniform number of background genes per gene. Note that left-multiplying a gene set matrix $\mathbf{W}^B$ by $\mathbf{G}$ maps the genes in the gene sets (columns of $\mathbf{W}^B$) to their corresponding background genes.

Then, given $\mathbf{G}$, for each cell $1 \le k \le C$ and module gene set $1 \le j \le M$,

$$Score_A\left(\mathbf{L}, \mathbf{W}^B\right)_{k,j} = \frac{\left(\mathbf{L}\mathbf{W}^B\right)_{k,j}}{\sum_{i=1}^{N}\mathbf{W}^B_{i,j}} - \frac{\left(\mathbf{L}\mathbf{G}\mathbf{W}^B\right)_{k,j}}{\sum_{i=1}^{N}\left(\mathbf{G}\mathbf{W}^B\right)_{i,j}},$$

where the first term simply computes the average expression of selected gene set genes in each cell of input gene expression matrix $\mathbf{L}$, and the second term calculates the aggregated average expression of background genes within the same cells.

The two resulting enrichment score matrices are subsequently concatenated into a single $C \times 2M$ matrix $\mathbf{K}$:

$$\mathbf{K} = \begin{bmatrix} Score_U\left(\mathbf{T}, \mathbf{W}^B\right) & Score_A\left(\mathbf{L}, \mathbf{W}^B\right) \end{bmatrix}$$

To transfer these enrichment scores into comparable spaces, CytoTRACE 2 standardizes each score across cells, yielding $C \times 2M$ matrix $\mathbf{K}^{norm}$. This standardization, implemented via torch. nn.BatchNorm1d from PyTorch v.2.0.0 with affine = False, tracks the mean and variance of each score during training. Once trained, the model applies these learned values, rather than dataset-specific values, for standardization at inference.

**Integration of scores.** To convert the gene set enrichment scores to a single score per cell per GSBN module, the normalized scores $\mathbf{K}^{norm}$ are passed through a feedforward layer, termed the 'enrichment layer' in the CytoTRACE 2 core model, containing the associated length $2M$ gene set enrichment score weight vector $\bar{V}$ and yielding length $C$ potency category score vector $\bar{q}$. As part of this process, dropout is applied to reduce overfitting during model training, with a predetermined fraction of the normalized scores set at random to zero. From the weights in each $\bar{V}$, concatenated across potency categories into matrix $\mathbf{V}$, the directionality and importance of each gene set can be interpreted (see 'Interpretability' below).

The model then integrates across the potency category scores produced by each GSBN module, concatenating the potency category score vectors into $C \times G$ potency score matrix $\mathbf{Q}$. This procedure represents the shared output layer of the CytoTRACE 2 core model.

To convert the logit entries of $\mathbf{Q}$ to likelihoods, the model applies a softmax activation function, yielding $C \times G$ matrix $\mathbf{P}$ representing the likelihood of each cell belonging to each of the six potency categories. The model then predicts cellular potency by assigning the potency category with highest likelihood for each cell, yielding length $C$ vector $\hat{\mathbf{y}}$:

$$\hat{\mathbf{y}}_k = \text{argmax}_{\{p\}_{p=1}^{G}}\left(\mathbf{P}_{k,*}\right)$$

The $\hat{\mathbf{y}}$ vector represents one of the key outputs of the CytoTRACE 2 core model; however, the model also computes an absolute developmental potential from this set of likelihoods, termed the raw potency score $R\bar{P}S$. For this aspect, we introduce length $G$ ordered vector $\vec{t}$ to be multiplied by the potency category likelihood matrix:

$$R\bar{P}S = \mathbf{P}\vec{t}$$

$$\vec{t} = [0.0, 0.2, 0.4, 0.6, 0.8, 1.0],$$

where $R\bar{P}S$ is the length $C$ raw potency score vector. As the potency categories are ordered based on their absolute developmental potential, the resulting raw potency score will be closer to one for higher potency categories, such as totipotent, and closer to zero for lower potency categories, such as differentiated. As $R\bar{P}S$ directly incorporates model uncertainty, it is passed to 'Postprocessing' below to define a more granular developmental ordering.

**Postprocessing.** As the fully trained CytoTRACE 2 model predicts potency for each cell individually, CytoTRACE 2 further processes the output (raw potency score $R\bar{P}S$ and predicted potency categories $\hat{\mathbf{y}}$) to incorporate the neighborhood structure of transcriptionally similar cells. We reasoned that doing so could further improve performance given our previous experience combining gene counts with transcriptional covariance in CytoTRACE 1 (ref. [3]). To this end, we devised and validated a three-step procedure using the training cohort, as described below. Notably, this procedure improves correlations with relative developmental orderings (see 'Metrics' below) over $R\bar{P}S$ or $\hat{\mathbf{y}}$ alone without sacrificing the potency classification performance achieved by $\hat{\mathbf{y}}$ (Extended Data Fig. 1b).

In the first step, CytoTRACE 2 applies Markov diffusion to smooth $R\bar{P}S$ using the same implementation as CytoTRACE 1 (ref. [3]). In brief, the $\log_2$-adjusted CPM/TPM gene expression input $\mathbf{L}$ is used to create

a Markov matrix from the transcriptional similarity between cells over the top 1,000 genes with highest dispersion[3]. This similarity matrix is then used to smooth $R\bar{P}S$ with diffusion parameter $\alpha = 0.9$ as previously described[3], yielding smoothed potency score $S\bar{P}S$. Using the same sampling procedure described in our previous work[3], the running time of this step can be significantly reduced without loss of performance (Extended Data Fig. 1c). In this study, sampling was restricted to datasets with >10,000 cells (Supplementary Table 1).

To reconcile $S\bar{P}S$ with predicted potency categories $\hat{\boldsymbol{y}}$, in the second step CytoTRACE 2 performs a binning procedure to maintain $\hat{\boldsymbol{y}}$ while preserving relative potency ordering within each category. To do so, CytoTRACE 2 first separates cells by their predicted potency category and assigns each cell $1 \le w \le C$ a rank $\mathcal{R}(k, \hat{\boldsymbol{y}}_w)$ relative to all cells sharing predicted potency category $\hat{\boldsymbol{y}}_w$. For this transformation, within each potency category $1 \le p \le G$, the cell with lowest potency score receives rank 1 while the cell with highest potency score receives maximum rank $r_{\max}(p)$. Cells are then arranged uniformly by rank per potency category within equal length partitions of the unit interval, yielding binned smooth potency score $S\bar{P}S^B$. Thus, the binned smooth potency score for differentiated cells extends from 0 to 1/6, unipotent from 1/6 to 2/6, and so on, with relative ordering within each bin matching that of the original smoothed potency score.

In the third step, to further smooth $S\bar{P}S^B$ while minimizing the impact on $\hat{\boldsymbol{y}}$ and allowing for the preservation of rare cell states (Extended Data Fig. 3f), CytoTRACE 2 applies a variation of $k$-nearest neighbor ($k$-NN) smoothing to datasets with >100 cells. Here, we introduce an efficient heuristic approach for adaptive neighborhood smoothing guided by two key assumptions: (1) cells with more similar gene expression profiles are more likely to share a potency phenotype; and (2) prediction errors for cells with the same ground truth potency exhibit a random distribution around a central mean. To balance these two considerations and identify an appropriate neighborhood size, we select $k$ adaptively for each cell according to the following process. First, given $\log_2$-adjusted CPM/TPM gene expression profiles for the selected cell, we standardize expression per cell to zero mean and unit variance, then perform dimension reduction of standardized gene expression profiles over all cells to the top 30 principal components (PCs). Using the top 30 PCs, we then compute pairwise Euclidean distances for all cells, rescaling the resulting distances to unit maximum per cell of interest. Next, we define the neighborhood around each center cell $w$ through an iterative procedure, allowing a maximum neighborhood size of 30 cells. We start with the nearest cell to $w$, denoted $c_1$, and calculate the average potency score prediction for $w$ and $c_1$, mapping the result to one of six broad potency categories, yielding $P_1$. We repeat this calculation for the next two nearest cells to $w$ ($c_2$ and $c_3$), yielding $P_2$, and compare $P_1$ and $P_2$. If identical, we assume that we have sufficiently captured the neighborhood, setting $k = 3$ (for the three non-self-neighbors) and exiting the process. If not identical, we repeat the procedure increasing the group size by one, in other words, comparing the nearest two cells to $w$ (yielding three total cells) with the next nearest three cells ($c_3$, $c_4$ and $c_5$). We repeat this process until the resulting potency categories are the same between two groups, in which case we select $k$ to encompass all cells considered between the two groups, or until we exhaust our candidate nearest neighbor cells (reach a group size of 15). If concordance between nearest and next nearest groups is not found, we keep our initial selection of $k = 3$.

Once $k$ is determined, we update our prediction for $w$ according to the distance-weighted mean of neighborhood potencies to obtain the final potency score prediction:

$$\overline{\text{cytotrace2}}_w = \frac{\sum\limits_{c \in N(w)} S\bar{P}S^B_c (1 - d_c)^2}{\sum\limits_{c \in N(w)} (1 - d_c)^2},$$

where $N(w)$ denotes the set of all cells within the selected neighborhood of center cell $w$, including $w$ itself, and $d_c$ denotes the Euclidean distance of cell $c$ to cell $w$. Categorical potency predictions are updated based on the defined intervals above, yielding $\hat{\boldsymbol{y}}^*$.

We found empirically that combining these three approaches yielded superior performance on the training cohort (Extended Data Fig. 1b).

## Training and hyperparameter tuning

**Loss function.** For model training, we defined a loss function combining cross-entropy loss with an additional term penalizing gene set size based on the binary weighting matrix $\mathbf{W}^B_p$ originating from each GSBN module, $1 \le p \le G$. More precisely, we define the loss function as the sum of gene set size penalty loss $J_S$ and a prediction loss per cell $J_P$:

$$J = J_S\left(\mathbf{W}^B_1, \cdots, \mathbf{W}^B_G\right) + \sum_w J_P\left(\hat{\boldsymbol{y}}_w, \boldsymbol{y}_w\right)$$

In detail, given potency category predictions $\hat{\boldsymbol{y}}_w$ and ground truth potency categories $\boldsymbol{y}_w$ for cell $w$ (see 'Single-cell potency atlas' above), we defined prediction loss $J_P$ as:

$$J_P\left(\hat{\boldsymbol{y}}_w, \boldsymbol{y}_w\right) = \bar{\boldsymbol{v}}_w \times CE\left(\hat{\boldsymbol{y}}_w, \boldsymbol{y}_w\right)$$

where $\bar{\boldsymbol{v}}_w$ denotes the loss weight assigned to cell $w$, and $CE(\hat{\boldsymbol{y}}_w, \boldsymbol{y}_w)$ denotes the cross-entropy loss for cell $w$. Loss weights for all cells are contained in the length $C$ weighting vector $\bar{\boldsymbol{v}}$, which has unit sum and is constructed hierarchically to assign equal weight (1) to all broad potency categories, (2) to all phenotypes within each broad potency category, and (3) to all datasets contributing to each phenotype.

We defined gene set size penalty loss $J_S$ as:

$$J_S\left(\mathbf{W}^B_1, \cdots, \mathbf{W}^B_G\right) = a\lambda \sum_{p=1}^{G} \left| \frac{1}{N}\left(\mathbf{W}^B_p\right)^T \left(\mathbf{W}^B_p\right) \odot \mathbf{I} \right|_F, \, a = \frac{\sqrt{12}}{\sqrt{M}}$$

where $|\cdot|_F$ denotes the Frobenius norm, $\odot$ denotes the Hadamard (or element-wise) product, $\mathbf{I}$ denotes the $M \times M$ identity matrix, $\lambda$ denotes the gene set size penalty weight, and $a$ serves as a scaling factor to make $J_S$ invariant to the number of gene sets included in $\mathbf{W}^B_p$, with factor $\sqrt{12}$ selected to anchor the gene set size penalty weight to the center of the range of hidden sizes tested (see 'Hyperparameter optimization'). This loss component serves to minimize the number of genes in each gene set while regularizing the training of the model.

**Model regularization.** To promote model generalizability, we introduced two explicit regularization aspects. We included a dropout layer to avoid model overfitting to specific enrichment scores ("Integration of scores"). A dropout layer[51] randomly drops (sets to zero) units in a hidden layer of a neural network. This layer was applied to the normalized scores $\mathbf{K}^{\text{norm}}$ during training only. Additionally, a penalty term was added to the loss function to constrain the number of genes in each gene set of $\mathbf{W}^B$ ("Loss function").

**Model initialization and updates.** Model weights were initialized according to PyTorch v.2.0.0 default except for the binary weighting matrices, which were initialized at random with values sampled from the Gaussian distribution with mean of $-0.1$ and s.d. of $0.055$ to produce a sparse initial binarization with approximately 500 genes selected per gene set.

Model training was performed with mini-batch learning using a batch size of 1,024. To balance batches and ensure equal representation for the model learning process, each batch was constructed via uniform sampling across datasets and phenotypes (Supplementary Tables 1 and 3) as implemented by torch.utils.data.WeightedRandomSampler in PyTorch.

Following initialization, forward propagation proceeded for each iteration as described in 'Core model architecture', with parameters updated according to their definition. For numeric stability, the cutoff rank $\tau$ ('Gene set binary networks') for trimming input rank space expression matrix **R** was not learned directly but rather computed as a function of learnable parameter $\tau_m \in \mathbb{R}$, which was initialized uniformly at random from $0 \leq \tau_m \leq 1$ per module and suitably scaled. As gene set enrichment score calculation ('Enrichment assessment') requires a gene set pool larger than the gene set itself for comparison, $\tau$ was computed from $\tau_m$ in such a way as to ensure that the ranks of at least ten more genes beyond the maximum gene set size of the module were preserved following trimming to **T**. Thus, at each iteration, the updated $\tau_m$ was scaled and constrained as follows:

$$\tau = 10 + \max_{1 \leq j \leq M} \bar{S}_j + 1,000 \times \max(0, \tau_m)$$

Model predictions were assessed at each iteration against ground truth, with the loss function and its gradient computed and used to backpropagate updates to network weights using PyTorch's NAdam optimizer with custom learning rate lr = 0.001 (see 'Hyperparameter optimization' below) and otherwise default parameters. Given the role of inertia in successfully training binary neural networks[52,53], we employed cross-epoch gradient accumulation to dampen binary weight flipping and achieve a stabilizing effect. This approach additionally facilitates broader hyperparameter space exploration while validation-based early stopping (see 'Model evaluation and stopping') ensures that the most performant model encountered during training is retained. Backpropagation for the binary neural network component of each GSBN module was implemented with Straight-Through Estimator and hardtanh activation function as previously described[48].

**Model evaluation and stopping.** We evaluated model validation performance via weighted accuracy, defined as the mean F1 score across evaluable potency categories. To do this, we first calculated the F1 score for each phenotype (standardized as in Supplementary Table 3) and dataset pair using metrics.precision_recall_fscore_support from sklearn v.1.0.2. We then averaged the resulting scores across datasets per phenotype, across phenotypes within each broad potency category, and across broad potency categories, yielding the final weighted accuracy. For the standard CytoTRACE 2 model, each validation set consisted of a single dataset; however, for the leave-clade-out model (see 'Generalizability to unseen cell-type clades'), validation sets included all cells covering a clade, regardless of dataset. All models were trained for 100 epochs with the best model weights by the highest score on the validation set after a minimum of 15 initial training epochs preserved and returned for the final model.

**Hyperparameter optimization.** To evaluate the hyperparameter space of CytoTRACE 2, we performed a hyperparameter sweep over the training cohort using wandb (v.0.16.4) (https://wandb.ai). We explored the learning rate lr over {0.01, 0.005, 0.001, 0.0005, 0.0001}, number $M$ of gene sets per broad potency category over {1, 2, 4, 8, 12, 16, 24, 32, 48}, gene set size penalty weight $\lambda$ over {0.5, 0.1, 0.05, 0.01, 0.005, 0.001}, dropout rate $\rho$ over {0, 0.25, 0.5}, and enrichment considering whether to use AMS enrichment, UCell enrichment, or the combination of both as described in 'Enrichment assessment' above. For every iteration of leave-one-dataset-out nested cross-validation, we trained models across 500 different combinations of these hyperparameters sampled based on the random hyperparameter search. To minimize overfitting to training data, we used a nested cross-validation framework. While one dataset was held out from training and evaluated as a validation set, another dataset was also held out from training but used to determine the early stopping point as described in 'Model evaluation and stopping'. We scored each hyperparameter combination by weighted accuracy over model validation sets (Supplementary Table 3; see 'Model evaluation and stopping').

We observed that variation in hyperparameter values had minimal impact on performance, underscoring overall model robustness (Extended Data Fig. 1e, left and Supplementary Table 6). Final hyperparameter selection was carried out by a manual curation process identifying values yielding consistently (albeit modestly) higher weighted accuracy. In selecting the number of gene sets $M$ per potency category, we found that model performance increased with $M$ before plateauing (Extended Data Fig. 1e, right); as such, we selected $M$ slightly larger than the number corresponding to the elbow of this curve. The final hyperparameters used were $M = 24$ gene sets per potency; $\rho = 0.5$ dropout probability; $\lambda = 0.01$ gene set size penalty weight; and lr = 0.001 learning rate.

Next, we evaluated the enrichment metrics. Among all models, we limited to 84 models with hyperparameter values in ranges of plateau ($M \geq 2$ gene sets per potency; $\rho = 0.5$ dropout probability, $\lambda \leq 0.01$, lr $\leq 0.001$). AMS enrichment and both AMS and UCell enrichment achieved superior performance compared to UCell enrichment alone (Extended Data Fig. 1f and Supplementary Table 6). Given the potential to enhance generalizability, we therefore selected the combination of AMS and UCell enrichment metrics for the final model.

**Model ensembling.** Models were trained via leave-one-dataset-out cross-validation for each of the training datasets, with final CytoTRACE 2 predictions in non-training data obtained as the result of integrating predictions across the 19 resulting models followed by an additional postprocessing step. As described in 'Integration of scores' above, each model $m$ yields a $C \times G$ potency category likelihood matrix $\mathbf{P}^m$. Models were integrated by entry-wise averaging of potency category likelihood matrices to yield a single potency category likelihood matrix $\mathbf{P}^{ensemble}$ from which potency category predictions and raw potency scores were computed as described above, before passing them to 'Postprocessing'.

## Dictionary of input genes

To create dictionary $\mathbb{D}$ ('Preprocessing' above), all human gene symbols were mapped to their closest mouse orthologs, as determined by gene sequence similarity, using the GRCh38.p13 and GRCm39 annotation files available from Ensembl v.109, respectively. In cases where a single mouse gene $g$ was identified as the best hit for multiple human genes, the human gene with maximum sequence similarity to $g$ was selected and the remaining human gene(s) excluded from further consideration. Unique human gene symbols without orthologs by the above process were also included for completeness. To define a common subset, only genes present in at least 80% of datasets from an initial development cohort, a subset of the final training cohort, were retained. Combining these steps, $\mathbb{D}$ was assembled with 14,271 unique gene symbols, including 13,750 orthologous pairs and 521 genes without orthologs in Ensembl via the mapping step above. When mapping human datasets to $\mathbb{D}$, gene symbol aliases are resolved using linked aliases available from https://biomart.genenames.org. When mapping to mouse datasets, alias gene symbols are resolved using data available from https://www.informatics.jax.org/mgihome/nomen/.

## Interpretability

The GSBN architecture of CytoTRACE 2 enables direct interrogation of the binary weight matrices, consisting of gene sets associated with each potency category (Fig. 1c and Extended Data Fig. 1a). By examining the orientation of the output layer weights for each gene set, we found that gene sets with positive weights (polarity) were highly enriched in a given potency category, whereas those with negative weights (polarity) were preferentially depleted (Fig. 2c). Additionally, we reasoned that genes repeatedly selected for a given potency category were more likely to be important for effective classification. As such, we designed a metric to quantify feature importance, assigning

importance scores to genes according to the frequency at which they were selected in positively versus negatively weighted gene sets. Here, we incorporate gene selection frequency across all 19 training models computed by leave-one-out cross-validation (LOOCV) over the training cohort datasets.

More formally, we define $N \times G$ feature importance score matrix $\mathbf{F}$ (Supplementary Table 15) containing the feature importance score of each gene $1 \leq i \leq N$ for each potency category $1 \leq p \leq G$ based on the gene set compositions and enrichment weights across models. Two enrichment weights correspond with each gene set, one per enrichment score type (see 'Enrichment assessment'). Given gene set enrichment weight matrix $\mathbf{V}^l$ of model $l$, we calculate the polarity Polarity $(\mathbf{V}^l, j, p)$ of gene set $j$ defined within model $l$ for potency category module $p$ as the sign of the average of these two weights. Then, relying on model binary weighting matrices to encode gene set composition, we construct feature importance score matrix $\mathbf{F}$ entry-wise as

$$\mathbf{F}_{i,p} = \sum_{l=1}^{19} \sum_{j=1}^{M} \mathbf{W}_{p,l}^{B}[i,j] \times \text{Polarity}\left(\mathbf{V}^l, j, p\right),$$

where $\mathbf{W}_{p,l}^{B}[i,j]$ denotes the $[i,j]th$ entry of the binary weighting matrix from module $p$ of model $l$.

## Performance assessment

**Metrics.** Two key metrics, illustrated in Extended Data Fig. 1d, were used to quantify reconstruction of known developmental orderings: absolute order and relative order. Absolute order quantifies cross-dataset performance, whereby predicted orderings from all cells with annotated potency levels are analyzed together, regardless of dataset, tissue type or platform (Supplementary Tables 2 and 3). Relative order quantifies performance within a given dataset and tissue type, akin to conventional pseudotime and ranges from 1 (least differentiated) to $N$ (most differentiated) in each dataset (Supplementary Table 4). For both metrics, we applied weighted Kendall correlation ($\tau$) (wdm package v.0.2.4 in R) to assess concordance between known and predicted developmental orderings, with weighting schemes provided in Supplementary Table 5. Similar to our previous work[3], ground truth phenotypes corresponding to less mature cells were coded with lower ranks (starting at 1); therefore, higher predictions of developmental order were ranked such that higher values received lower ranks and vice versa.

For categorical predictions (CytoTRACE 2 and potency classification benchmarking outputs only), we evaluated potency classification performance as well. Binary correctness of predicted versus ground truth broad potency categories was assessed via mean multiclass F1 score, implemented with function f1_score from sklearn.metrics with average = none (Extended Data Figs. 1c top, 2d second from right, 3b–e left bottom, 7a left and 7b $x$ axis). To account for the magnitude of deviations from ground truth potency, we also considered mean absolute error (MAE), assigning each broad potency class an integer label corresponding to the class ordering, with labels ranging from 1 (differentiated) to 6 (totipotent), and computing the absolute value of the difference between predicted and ground truth categories (Extended Data Figs. 2d far right, 3b–e right bottom, 7a right and 7b $y$ axis). For both metrics, scores were computed per ground truth potency category then aggregated by mean across potencies.

**Generalizability to unseen cell-type clades.** To test the generalizability of CytoTRACE 2 to unseen developmental systems, we trained a version with a leave-clade-out framework (Fig. 1f), grouping phenotypes into 18 mutually exclusive developmental clades as detailed in Fig. 1b and Supplementary Table 9. Of note, to ensure representation of some totipotent and pluripotent phenotypes for all training sets, we partitioned embryonic phenotypes into two clades by alternating granular potency level annotation, corresponding to distinct time points during development and resulting in 19 total clades for this

analysis (Supplementary Table 2). The final clades cleanly separate, for example, immune cells, neural cells, endothelial cells, connective tissue cells and bone cells, among others. Stem and progenitor cells that produce a given clade were included in the same partition as that clade (for example, pancreatic multipotent progenitors were included with pancreatic epithelial cells). Epithelial cells were separated by tissue to avoid conflating tissue-specific developmental hierarchies. For each clade, we trained an ensemble of two models over the remaining 18 clades, selecting at random 17 clades for training and one clade as a held-out validation set to be used for early stopping (see 'Model evaluation and stopping') for each model. We then applied the resulting ensemble to the unseen test clade, assessing performance across all held-out clades in Fig. 1f.

**Randomization of training and test sets.** To assess the robustness of the model to variation in the composition of the training cohort, we repeated the CytoTRACE 2 training process as described in 'The CytoTRACE 2 framework' across a series of three randomized splits covering all 33 datasets in the single-cell potency atlas (Supplementary Table 8). We partitioned the datasets at random into three folds, each containing 11 datasets. To ensure minimum adequate representation within each category, we confirmed that each fold contained at least one phenotype per broad potency category. Tabula Muris, which was divided into two sub-datasets according to platform for the original CytoTRACE 2 training cohort due to its size and diversity, was again divided, with one of its sub-datasets assigned to another fold at random. For each split, two folds were combined to form the training cohort and the remaining one left as a test set for evaluation (2:1 training–test split; Supplementary Table 8). Performance per test set of these three randomized splits, along with the original CytoTRACE 2 test set, was assessed by absolute order, relative order, mean multiclass F1 score and MAE (see 'Metrics'), showing strong consistency across folds (Extended Data Fig. 2d). Performance for the three randomized splits was additionally assessed across all held-out datasets jointly in Extended Data Fig. 2e.

## Robustness of CytoTRACE 2

**Robustness to annotation error.** To evaluate the robustness of CytoTRACE 2 to potential noise within potency annotations, we trained models across two scenarios of training cohort annotation error, then evaluated model performance over the test cohort (see 'Training and test datasets'). To simulate annotation error, we formulated label noise as a transition matrix[54], encoding the probability of perturbation from one potency to another (Extended Data Fig. 3a). Transition matrix perturbation probabilities were designed to follow a Gaussian distribution based on the rank distance between the original potency and perturbed potency. In detail, the probability that the potency label of cell $s$ transitions from true potency $j$ to perturbed potency $i$

$$P(s_i | s_j) = \frac{1}{\sqrt{2\pi\sigma^2}} \exp\left(-\frac{(j-i)^2}{2\sigma^2}\right), i, j \in \{1, 2, 3, 4, 5, 6\}$$

where potencies $i, j$ are represented by their rank within the six broad potency categories. The s.d. values ($\sigma$) were selected to yield a titration of 5%, 10%, 20%, 50% and 80% perturbation levels. Rows were normalized to unit sum for a net probability of one. For the first annotation error scenario, we considered cell-level annotation error and perturbed the potency annotations of individual cells independently (Extended Data Fig. 3b). For the second, we considered phenotype-level annotation errors and simultaneously perturbed the potency annotations of the entire standardized phenotypes (Extended Data Fig. 3c).

**Robustness to variation in gene counts and UMI counts.** To determine the influence of variable gene counts and unique molecular identifier (UMI) counts on CytoTRACE 2, we performed two experiments

in which scRNA-seq expression data from all 14 datasets in the test cohort were perturbed by downsampling gene counts (Extended Data Fig. 3d) and all seven droplet-based datasets in the test cohort (Supplementary Table 1) were perturbed by downsampling UMIs (Extended Data Fig. 3e). We assessed the robustness of the model to different gene counts by downsampling the expression data of each cell to the same number of genes: 2,000, 1,000, 750, 500, 250 and 100. We selected the top genes by highest expression and set the expression of the remaining genes to zero. For any expression level ties at the threshold, we selected the genes to include to reach the target gene count at random. The downsampling process for UMIs consisted of randomly sampling the expression data of each cell based on the transcriptome probability distribution, defined as the fractional expression of each gene after scaling the sum of UMIs in each cell to one. Then, using the raw count matrices, we downsampled the expression data of each cell to the same number of UMIs: 5,000, 3,000, 2,000, 1,000, 500 or 100 UMIs. Cells with UMIs lower than a given threshold were unaltered. We repeated each process for five replicates, then assessed performance for standard metrics as described above (see 'Metrics') relative to the CytoTRACE 2 predictions without perturbation.

**Robustness to titration of cell type rarity.** Given the inclusion of neighborhood-based smoothing in model postprocessing, we performed a titration experiment applying CytoTRACE 2 to test datasets with selected phenotypes downsampled to increasingly rare abundance. For 11 phenotypes spanning a range of potencies, we downsampled cells of the selected phenotype to predefined abundances of 50, 20, 10, 8, 5, 2 and 1 cell(s), leaving the remaining cells in the dataset unchanged. We repeated this titration process five times for each phenotype, observing robust predictions down to five cells per phenotype (Extended Data Fig. 3f). As such, we recommend that the final postprocessing step (adaptive $k$-NN smoothing) be omitted when exceedingly rare cell states (consisting of <5 cells each) are of interest.

**Analysis of mouse embryogenesis**
For the analyses presented in Extended Data Fig. 5, we downloaded and curated six publicly available scRNA-seq datasets spanning each embryonic day during mouse prenatal development[2,21–25] (Supplementary Table 1). One dataset, which covers pre-implantation through early implantation (E0.5–E4.5) (Deng et al.[22]), was obtained from the 19-dataset training cohort (Supplementary Table 1) and evaluated using a CytoTRACE 2 model trained on the remaining 18 datasets to avoid overfitting (see 'Benchmarking developmental potential inference methods and annotated gene sets'). Four datasets[21,23–25] covering embryogenesis periods from implantation to organogenesis were previously assembled by Qiu et al.[25] and are accessible through http://tome.gs.washington.edu. Finally, a single-nucleus RNA-seq dataset[2] covering organogenesis through birth (E8.75–P0) and generated by sci-RNA-seq3 was downloaded from http://mouse.gs.washington.edu. As we compared CytoTRACE 2 against multiple methods with highly variable time complexity ('Benchmarking developmental potential inference methods and annotated gene sets'), all cells were randomly downsampled to 30 cells per author-supplied phenotype per time point, resulting in a combined dataset of 183,771 cells. This allowed us to balance considerations of performance versus computational efficiency. We ran each method on each dataset individually as described in 'Benchmarking developmental potential inference methods and annotated gene sets'. No dataset integration or batch normalization procedures were applied. For Organogenesis (E8.5)[25] and Organogenesis (E8.5–P0)[2], which were sequenced using sci-RNA-seq3, we used count data after running SCTransform of Seurat (v.4.3.0) with default parameters. Due to the large size of the dataset, Organogenesis (E8.75–P0)[2] was run with ten randomly divided batches for SCENT (SR) and SLICE. Primordial germ cells were excluded owing to the wide range of potency levels reported in previous literature[55].

For the analyses in Extended Data Fig. 5d,e, we leveraged a data-driven lineage tree of mouse embryogenesis encoded as a directed acyclic graph[2]. Although the tree was constructed using a heuristic approach based on transcriptional covariance across embryonic time, it reflects many known parent-daughter relationships[2]. It thus serves as a proxy for developmental potential. We defined ground truth as the distance from the root (zygote) to each daughter node (Extended Data Fig. 5d, top). Using matching phenotype labels between the tree and the data presented in Extended Data Fig. 5a, CytoTRACE 2 potency scores were averaged by phenotype, balanced first by time points within a given embryonic day (if any) and then by embryonic day. If the same phenotype was present in more than one dataset, we weighted equally by dataset. For each direct path in the tree (from root to leaf), the resulting scores were then converted to rank space (Extended Data Fig. 5d, center). To reconcile cases where a given node $i$ participates in multiple paths, we used the average rank for $i$. CytoTRACE 1 predictions were processed in the same manner (Extended Data Fig. 5d, bottom). The resulting ranks were correlated with ground truth distances (distance from the root) in Extended Data Fig. 5e.

**Application to cancer types with known developmental states**
**Acute myeloid leukemia analysis.** For the analysis presented in Extended Data Fig. 6a, we downloaded the Galen et al.[56] acute myeloid leukemia (AML) dataset (Gene Expression Omnibus (GEO) accession number GSE116256; PMID 30827681) from the Curated Cancer Cell Atlas website on 28 June 2023 (https://www.weizmann.ac.il/sites/3CA/)[57]. We leveraged author-supplied cell type annotations, including classifications of malignant and non-malignant cells from 3CA[57]. From this dataset, comprising 28 samples with malignant cells, we excluded two cell line samples ('MUTZ3' and 'OCI-AML3'). We ran CytoTRACE 2 with default parameters ('Benchmarking developmental potential inference methods and annotated gene sets') on all annotated malignant cells from each tumor sample. For quality control, we further excluded samples for which each predicted potency label contained <10 malignant cells. For each of the resulting tumor samples ($n = 19$), we created a single matrix of malignant cells and non-malignant cells, with the latter uniformly downsampled from all patients to 100 cells per author-supplied phenotype ('B_cell', 'erythrocyte', 'myeloid', 'NK_cell', 'plasma' and 'T_cell'; non-malignant cells labeled as 'undifferentiated' were excluded from additional analysis). We then calculated the $\log_2$ fold changes (LFCs) of each potency category versus all other phenotypes by tumor sample and averaged by potency category across tumor samples. Finally, we normalized the logFC values of each gene to mean zero and unit variance across potency categories and plotted the enrichment of AML cell-type-specific gene signatures[26] ('LSPC-Primed-Top100', 'LSPC-Quiescent', 'GMP-like-Top100' and 'Mono-like-Top100'; https://github.com/andygxzeng/AMLHierarchies), each expected to be enriched in multipotent, multipotent, oligopotent and unipotent/differentiated cells, respectively (Extended Data Fig. 6a and Supplementary Table 10).

**Oligodendroglioma analysis.** For Extended Data Fig. 6b, we applied CytoTRACE 2 to scRNA-seq profiles of six oligodendrogliomas[27], with coordinates for the associated oligodendroglioma 2D lineage hierarchy embedding obtained from https://singlecell.broadinstitute.org/single_cell/study/SCP12/oligodendroglioma-intra-tumor-heterogeneity. We then assigned malignant oligodendroglioma cells to four transcriptional states following the protocol described by the authors[27] and visualized the association of CytoTRACE 2 potency predictions with the author-supplied stemness score. For the latter, we separated cells according to the stemness score by partitioning them into successive intervals of 0.25 units. We then displayed CytoTRACE 2 potency scores as a function of each interval (Extended Data Fig. 6b, right).

## Benchmarking cell type prediction methods adapted for potency classification

To evaluate CytoTRACE 2 against supervised machine learning approaches commonly employed in cell type prediction tasks (Extended Data Fig. 7a,b), we selected three dedicated single-cell annotation methods with superior performance in a benchmarking study[28] (scPred[30], SingleCellNet[31] and scmap[32]) and five general-purpose classifiers (below), each trained to predict six broad potency labels based on single-cell expression profiles.

All tools were trained and tested over a series of four folds, including the original CytoTRACE 2 training–test split (Fig. 1b) along with three randomized splits (see 'Randomization of training and test sets'), collectively encompassing all 33 ground truth datasets in the single-cell potency atlas described above, with classification performance per test cohort assessed by mean multiclass F1 score and MAE (Extended Data Fig. 7a and b; see 'Metrics'). For all methods, expression data were first mapped into the uniform feature space used by CytoTRACE 2 (see 'Preprocessing' and 'Dictionary of input genes'). Unless otherwise specified, and for all general-purpose classifiers, expression data were then CPM/TPM normalized and $\log_2$-transformed and subsequently standardized per cell to zero mean and unit variance. Other normalization schemes generally yielded worse performance and were thus omitted from further consideration ($\log_2$-adjusted CPM/TPM data, either used alone or with gene-level standardization). No explicit dataset integration or batch correction was performed. For general-purpose classifiers, versions were trained with and without sample weighting (computed as for CytoTRACE 2; see 'Loss function') for class imbalance mitigation, with the best performing version across all folds selected for each. All parameters were set to default values unless otherwise specified.

**CytoTRACE 2.** We applied CytoTRACE 2 with model ensembling and postprocessing as described in 'The CytoTRACE 2 framework' to predict cell potency categories. Datasets containing more than 100,000 cells were processed in batches of 100,000 cells, and diffusion was applied in batches of 10,000 cells for datasets exceeding 10,000 cells.

**scPred.** A dedicated cell type classification method, scPred first performs a dimension reduction, identifying PCs exhibiting significant variation across classes, then, as the default option, applies a support vector machine approach for classification[30]. Following the recommended pipeline for scPred (v.1.9.2) as described at https://powellgenomicslab.github.io/scPred/articles/introduction.html, we first normalized and scaled expression data using the NormalizeData() and ScaleData() functions in Seurat (v.5.1.0), respectively. We then used scPred's getFeatureSpace() function to identify class-informative PCs, trainModel() to train the default support vector machine (SVM) with radial kernel model for each potency category (one-versus-rest), and scPredict() for classification. A relaxed probability threshold of 0 was used to avoid 'unassigned' labels.

**SingleCellNet.** SingleCellNet performs cell type classification using a random forest multiclass classification approach[31]. Here, we trained the method over unnormalized expression data via the scn_train function of pySingleCellNet (v.0.1.1) with nTopGenes = 200, nTopGenePairs = 200, nRand = 100, nTrees = 1,000, stratify = False, and propOther = 0.4, following the tutorial provided at https://pysinglecellnet.readthedocs.io/en/latest/notebooks/train_classifier.html. The scn_classify() function with nrand = 0 was used for classification.

**scmap.** scmap uses a clustering approach to project cells onto a reference dataset for cell type classification[32]. Following the recommended pipeline for scmap (v.1.26.0) provided at https://bioconductor.org/packages/devel/bioc/vignettes/scmap/inst/doc/scmap.html, we $\log_2$-transformed expression data, then used selectFeatures() to select

informative genes and indexCell() to create a scmapCell index for the training dataset. For classification, we used scmapCell() to project the index onto the test dataset and scmapCell2Cluster() to obtain label assignments. A relaxed probability threshold of 0 was set to assign labels to as many cells as possible regardless of assignment confidence.

**Logistic regression.** We trained a logistic regression model to perform cell potency classification using the SGDClassifier from scikit-learn (v.1.4.2) with loss = 'log_loss', default L2 regularization, and sample weights provided for class balancing. This function internally employs a one-versus-rest (OVR) strategy, training a separate binary classifier for each potency category and selecting the potency category with highest confidence at evaluation.

**XGBoost.** We trained and applied the XGBClassifier function from the XGBoost library (v.2.1.1) with default parameters and without sample weights. Like logistic regression, this method uses the OVR approach.

**Linear SVM.** We implemented a linear SVM model using Scikit-learn's SGDClassifier with loss = 'hinge' for linear support vector classification with OVR. Sample weights were provided during training.

**Radial SVM.** We implemented an additional SVM version using SVC from scikit-learn (v.1.4.2) with the default radial basis function kernel and γ = 'auto'. The default decision function, which employs an inference of OVR from one-versus-one fits internally, was used. Sample weights were not provided during training.

**Multinomial logistic regression.** Using LogisticRegression from scikit-learn (v.1.4.2) with multi_class = 'multinomial', we fit a single logistic regression model for all potency categories simultaneously using cross-entropy loss and the 'sag' solver. A maximum number of iterations (max_iter = 500) and tolerance (tol = $1 \times 10^{-3}$) were set to ensure convergence. Sample weights were not provided during training.

## Benchmarking developmental potential inference methods and annotated gene sets

To rigorously assess performance on our compendium of 33 curated scRNA-seq datasets, we compared CytoTRACE 2 with eight published methods for predicting developmental potential from scRNA-seq data as well as nearly 19,000 previously annotated gene sets (Fig. 1h,i and Supplementary Tables 11–13). Unless otherwise stated, all evaluated methods and gene sets were applied to scRNA-seq datasets individually, without batch correction or integration across datasets, with expression data normalized per author recommendations and with default parameters. All expression data were subset to the cells with known potency. Each tissue and platform pair of Tabula Sapiens[33] and Tabula Muris[43] datasets were run separately.

Several methods rely on human gene symbols, as noted below. For all such instances, we mapped mouse dataset gene symbols to their closest human orthologs, as determined by gene sequence similarity, using the GRCm39 and GRCh38.p13 annotation files available from Ensembl, respectively. In cases where a single human gene $g$ was identified as the best hit for multiple mouse genes, the mouse gene with maximum sequence similarity to $g$ was selected.

As several methods have slower running times, to promote an equitable comparison while achieving computational feasibility, larger datasets were first downsampled. The Tabula Muris[43] dataset was downsampled to 30 cells per phenotype, separated by tissue and platform pair, and the 'Immune cell atlas (10x)', 'Human breast 1 (10x)', 'Human breast 2 (10x)', and Tabula Sapiens[33] datasets were downsampled to 100 cells per phenotype (Supplementary Table 1). Cell types in Tabula Sapiens[33] with fewer than five cells were removed after the prediction of each method to overcome the reduced data quality of Tabula Sapiens[33] ('Training and test datasets').

**CytoTRACE 2.** We applied CytoTRACE 2 with model ensembling and postprocessing as described in 'The CytoTRACE 2 framework' to predict cell potency categories and scores. Datasets containing more than 100,000 cells were processed in batches of 100,000 cells, and diffusion was applied in batches of 10,000 cells for datasets exceeding 10,000 cells. To evaluate the 19 scRNA-seq datasets included in the CytoTRACE 2 training cohort, we trained a separate model for each over the remaining 18 datasets. All other datasets were evaluated with the primary version of CytoTRACE 2 trained over all training datasets.

**CytoTRACE 1.** CytoTRACE 1, the predecessor of CytoTRACE 2, introduced transcriptional diversity quantified through gene counts as a correlate of developmental potential and exploited this concept to predict relative cellular potency from scRNA-seq[3]. CytoTRACE 1 (v.0.3.3) was applied with default parameters.

**SCENT (SR).** SCENT estimates relative cellular potency from scRNA-seq and a reference protein–protein interaction (PPI) network using single-cell signaling entropy (SR), a measure of the diversity of molecular pathway activity in a cell[15]. SCENT (v.1.0.3) was executed with the 'net13Jun12' human PPI network provided with the package and otherwise default parameters. For mouse datasets, genes were first mapped to human orthologs as described above. All gene symbols were converted to Entrez ID using org.Hs.eg.db (v.3.15.0) in R. Gene expression matrices were normalized per documentation recommendation (https://github.com/aet21/SCENT/blob/master/vignettes/SCENT.Rmd).

**SCENT (CCAT).** CCAT, implemented within the SCENT package, was developed as a highly efficient alternative to the original SCENT method, SCENT (SR)[14]. CCAT was applied with the same package, PPI network, and preprocessing steps described above ('SCENT (SR)') with expression datasets prepared as per documentation recommendations.

**FitDevo.** Similar to SCENT (CCAT), FitDevo infers cellular potency from the correlation between gene expression and a measure of gene weights[20]. FitDevo (v.1.2.0) was applied following tutorial instructions with binary gene weight matrix downloaded from the same source (https://github.com/jumphone/FitDevo/#demo-1–infer-developmental-potential-dp-using-expression-matrix-of-scrna-seq-data).

**SLICE.** SLICE relies on transcriptomic entropy for cellular potency prediction and lineage reconstruction, estimating entropy over functional groups of genes computed from Gene Ontology annotations[17]. SLICE (v.0.99.0) was applied according to demo details from the method's GitHub page (https://github.com/xu-lab/SLICE/blob/master/demo/FB.R).

**StemID.** StemID infers cellular differentiation trajectories from scRNA-seq data with a clustering-based algorithm analyzing links between clusters[16]. StemID, implemented in RaceID (v.0.1.4), was run according to documentation vignette instructions (https://cran.r-project.org/web/packages/RaceID/vignettes/RaceID.html). For each dataset, an SCseq object was initialized from each input gene expression matrix using filterData() with mintotal = 10. Ltree() and compentropy() were then applied consecutively to obtain the StemID score for cell potency.

**scTour.** scTour implements a deep learning architecture combining a variational autoencoder with a neural ordinary differential equation to reconstruct the developmental trajectory of an input scRNA-seq dataset, oriented according to gene counts[19]. scTour (v.1.0.0) was trained and applied to each dataset individually per 'Model training' documentation vignette instructions at https://sctour.readthedocs.io/en/latest/notebook/scTour_inference_PostInference_adjustment.html.

When the raw count matrix was available for the dataset, the negative binomial conditioned likelihood loss function was used. Otherwise, the CPM/TPM expression matrix was $\log_2$-transformed, and the mean squared error loss function was used instead. Cell potency scores were obtained from the developmental pseudotime predictions extracted from the model training output with get_time().

**mRNAsi.** mRNAsi utilizes a one-class logistic regression framework to construct a cellular stemness index applicable to cell potency estimation from bulk and scRNA-seq data[18]. mRNAsi was trained as described previously[3]. All input gene expression matrices were CPM/TPM normalized and $\log_2$-transformed.

**Gene sets.** The predictive capacity of 18,706 annotated gene sets (17,810 gene sets from MSigDB[36] and 896 gene sets of transcription factor binding sites from ENCODE/ChEA[34,35]) was assessed via GSEA. For each gene set, the AddModuleScore() function with default parameters from Seurat (v.4.3.0) was applied to each expression matrix normalized via Seurat's NormalizeData() function.

## Comparison to scVelo

As scVelo[5] relies on splicing kinetics, necessitating the processing of raw sequencing data, we limited our analyses to nine ground truth datasets from the test cohort that were generated by platforms with built-in support by velocyto and for which raw sequencing data are publicly available (Supplementary Tables 1 and 14). Raw FASTQ files for seven of these datasets, namely 'BM-MNC (CITE-seq)', 'Retinal neurons (10x)', 'Pancreas (10x)', 'Peripheral glia (Smart-seq2)', 'Skeletal stem cell (C1)' and 'HSCs and MPPs (inDrop)', were obtained from the Sequence Read Archive (SRA) from NCBI, with study IDs SRP188993, SRP168426, SRP200419, SRP109011, SRP239468 and SRP094420, respectively. For 'Peripheral glia (Smart-seq2)', we analyzed sample IDs prefixed with 'E12.5'. Notably, raw FASTQ files were only available for 227 of 473 cells in the 'Skeletal stem cell (C1)' dataset. For the remaining two datasets, 'Mouse neurogenesis (10x)' and 'Mouse mature neural cell types (10x)', data were obtained as BAM files from SRA study ID SRP476153.

FASTQ files were downloaded using sra-tools v.3.1.1 and processed with cutadapt v.4.9 for adaptor trimming of Smart-seq2/C1 reads. For preprocessing of inDrop samples, dropest v.0.8.6 was used (according to recommended workflow at https://velocyto.org/velocyto.py/tutorial/cli.html#run-dropest-run-on-dropseq-indrops-and-other-techniques). Reads were mapped and sorted BAM files were generated with STAR (v.2.7.11b) and Cell Ranger (v.8.0.1) using GRCm39 and GRCh38.p13 reference genomes for mouse and human datasets, respectively. Loom files containing spliced, unspliced and spanning reads were then generated from the BAM files along with corresponding Gene Transfer Format files using the velocyto.py v.0.17.17 Python command line tool.

Following quantification of spliced/unspliced counts, the scVelo v.0.3.1 Python velocity estimation workflow was run as described in the tutorial at https://scvelo.readthedocs.io/en/stable/. For all datasets, both a generalized dynamical model (as detailed at https://scvelo.readthedocs.io/en/stable/DynamicalModeling.html) and a differential kinetics adjusted model with grouping by the CytoTRACE 2 standardized phenotypes (as detailed at https://scvelo.readthedocs.io/en/stable/DifferentialKinetics.html) were employed. With the exception of random_state in scvelo.pp.neighbors(), which was set to 0 to ensure reproducible results, all other parameters were set to those in the respective vignettes, including min_shared_counts in scvelo.pp.filter_and_normalize(), which was set to 20 for dynamical models and 30 for differential kinetics models. Following velocity estimation, cell-internal latent time was inferred using scvelo.tl.latent_time(). The resulting outputs were then evaluated via absolute and relative order (see 'Performance assessment' above) and CytoTRACE 2 outputs were assessed over the same cells for comparison (Extended Data Fig. 8 and Supplementary Table 14).

## Analysis of potency-associated molecular programs

**Visualization and specificity of potency programs learned by CytoTRACE 2.** For the analyses presented in Fig. 2b and Extended Data Fig. 9a, we ran CytoTRACE 2 on each of the training and test datasets, then extracted positive potency score matrix $\mathbf{Q}^{pos}$ from each of the 19 models per dataset. $\mathbf{Q}^{pos}$ is derived from the final layer of each GSBN module; obtained similarly to the potency score matrix $\mathbf{Q}$ in 'Integration of scores' above, but by multiplying $\mathbf{K}^{norm}$ by only positive weights from the enrichment layer; and has a dimensionality of $C$ input cells by 114 when combined across the 19 models and six potency category modules. We concatenated $\mathbf{Q}^{pos}$ matrices from the 19 models across all 33 datasets ($C = 124{,}231$ cells; downsampled as described in 'Benchmarking developmental potential inference methods and annotated gene sets' above) to produce $\mathbf{Q}^{pos}_{all}$, standardized across all cells in the training and test sets separately. We fitted principal-component analysis (PCA) from scikit-learn (v.1.1.1) to the training set component of the resulting matrix, retaining the first three PCs, then applied the resulting projection to the training and test set components individually. Next, we repeated this process fitting Uniform Manifold Approximation and Projection (UMAP) from umap-learn (v.0.4.6) to the PCA projection of the training component, then applying the resulting UMAP projection to the PCA projection of the training and test components individually. To adjust for differences in cell density that confound visualization, we averaged CytoTRACE 2 potency scores within each window of 0.5 UMAP units squared across the two components of UMAP space. The same procedure was applied to visualize the ground truth potency of each cell (Fig. 2b, bottom).

**Top potency-associated genes learned by CytoTRACE 2.** For the analysis presented in Fig. 2c, we examined the expression of the top 500 potency-associated markers learned by CytoTRACE 2 (matrix $\mathbf{F}$ in 'Interpretability') in training and test sets from the single-cell potency atlas. We first filtered and mapped gene symbols in every dataset to CytoTRACE 2 input features ($n = 14{,}271$), then CPM/TPM normalized as appropriate and $\log_2$-adjusted the data. Keeping training and test data separate, the expression matrices from each dataset were mean-aggregated into pseudo-bulk expression profiles by phenotype. We then further averaged shared phenotypes across datasets profiled by the same general platform (droplet/UMI or plate-seq/non-UMI) and finally, by species identifier (human or mouse). This resulted in a $14{,}271 \times 237$ matrix, with 14,271 genes (rows) and 237 phenotype, species and platform combinations across training and test sets (columns). Using this matrix, we calculated the mean expression of the top 500 positive/negative genes per potency category (matrix $\mathbf{F}$ in 'Interpretability'; Supplementary Table 15), then unit variance normalized the resulting expression signatures across pseudo-bulk samples separately for training and test sets (Fig. 2c). See also Supplementary Table 16.

**Validation of CytoTRACE 2 by large-scale functional genomics.** To assess the biological relevance of CytoTRACE 2 model features, we analyzed large-scale in vivo CRISPR screening data of mouse hematopoiesis[38] (Fig. 2d). These data encompass ~7,000 genes along with CasTLE $-\log_{10} P$ values, representing the effect of knockout (KO) on HSC differentiation. Although these effects were separately measured for lymphoid ($n = 6{,}783$ genes) and myeloid ($n = 6{,}732$ genes) lineages, directed $-\log_{10} P$ values were well correlated between them ($r = 0.78$). Therefore, to create a single ordered list of KO effects, we combined directed $-\log_{10} P$ values for genes with effect score data in both lineages, keeping the most significant directed $-\log_{10} P$ value for each gene (positive or negative). Contributions from each lineage were nearly perfectly balanced, with higher positive scores and higher negative scores implying that KO of a gene promotes or inhibits HSC differentiation, respectively. We then intersected the resulting vector with those within the CytoTRACE 2 model space, resulting in $n = 5{,}757$ genes. We applied

fgsea (v.1.25.1) to the rank-ordered list to jointly evaluate the enrichment of the top 100 positive and negative multipotency markers from the CytoTRACE 2 feature matrix (Fig. 2e and Supplementary Table 15). $P$ values were computed using the adaptive multilevel Monte Carlo method and $Q$ values represent false discovery rates calculated using the Benjamini–Hochberg procedure.

To assess robustness across the number of top multipotency markers selected, we repeated the above process checking 50, 100, 200 and 500 markers (Extended Data Fig. 9c and Supplementary Table 15). We also compared the median $-\log_{10} Q$ value of gene set enrichment over the four gene set sizes against the same process repeated for CytoTRACE 2 markers for all other potency categories (Extended Data Fig. 9d and Supplementary Table 15).

**Functional annotation analysis.** To interpret potency-associated genes learned by CytoTRACE 2, we applied fgsea (v.1.25.1) to each rank-ordered gene list in $\mathbf{F}$ with minSize = 15 and otherwise default parameters. $\mathbf{F}$ is an $N \times G$ matrix consisting of model importance scores for all $N$ evaluable genes (14,271) in each of $G = 6$ potency categories learned on the training cohort ('Interpretability' above; Supplementary Table 15). Mouse and human MSigDb signatures from MH/H: hallmark gene sets, M2/C2: curated gene sets, including CGP and CP:WIKIPATHWAYS, CP:REACTOME and CP:KEGG_MEDICUS; and M5/C5: ontology gene sets were downloaded from https://www.gsea-msigdb.org/gsea/msigdb/. We ran fgsea on mouse and human gene sets separately, and human gene sets were limited to those with no counterpart in mouse gene sets. When running human gene sets, genes in $\mathbf{F}$ were first mapped to human orthologs by dictionary $\mathbb{D}$ ('Dictionary of input genes'). Gene sets with an adjusted $P$ value < 0.05 in at least one potency category are provided in Supplementary Table 17. We selected a subset of representative molecular signatures for display in Extended Data Fig. 9e, highlighting both canonical and poorly understood potency-related biology.

**Analysis of multipotency-associated programs.** All WikiPathways gene sets from canonical pathways (CP) in M2/C2 with positive normalized enrichment scores in multipotency (see 'Functional annotation analysis' above) are presented in Fig. 2f. Next, we assessed the gene set comprising the UFA factors *Fads1*, *Fads2* and *Scd2* (Fig. 2g) for specificity and conservation across tissues.

For this purpose, we analyzed pseudo-bulk-expression profiles of each phenotyp–dataset pair in our 33-dataset potency atlas using single-sample GSEA (ssGSEA) from the GSVA package in R (v.1.46.0)[58] to mitigate technical variation. Once ssGSEA scores were obtained for the UFA factors, we then averaged them into the same 237 phenotype, species, and platform combinations described in 'Top potency-associated genes learned by CytoTRACE 2'. Keeping training and test cohorts separate, we further averaged ssGSEA scores by developmental system (here, denoted 'tissue'), using the phenotype-to-clade mapping scheme provided in Supplementary Table 9. Mean-aggregated ssGSEA scores across 237 phenotype, species and platform combinations in training and test sets are displayed in Fig. 2h.

Statistical assessment of the specificity of UFA genes to multipotency was performed via permutation testing. First, we took the median value of the ssGSEA scores in each ground truth potency category. Next, we calculated the pairwise difference $\Delta_i$ between the median ssGSEA scores of multipotent and each other potency category $i$. We then calculated two test statistics: $\min(\Delta_i)$ and $\text{mean}(\Delta_i)$. To simulate a null distribution, we permuted the phenotype-level ssGSEA scores, recomputed the median ssGSEA score for each ground truth potency category and calculated both statistics. We repeated this process 10,000 times. To determine an empirical $P$ value, we tallied the proportion of times both statistics were as high (or higher) than the test statistics from the original data. We did this for the multipotent category separately across the training and test cohorts (Fig. 2h).

AUCs of UFA genes (main text) were calculated for training and test sets separately using the ssGSEA scores described above, but after averaging the scores by tissue type to address imbalances. AUCs were first calculated in a pairwise manner between multipotency and each other potency category, then averaged.

### Experimental validation of UFA genes in multipotency

**Mice.** C57BL/6 mice were purchased from The Jackson Laboratory and housed in the Stanford Animal Facility. For all analyses shown in Fig. 2i,j and Extended Data Fig. 10, 8–12-week-old mice were used, with equal numbers of males and females. Mice were maintained in-house under aseptic sterile conditions and supplied with autoclaved food and water. The animals were housed under a 12-h light–dark cycle at room temperatures between 20–26 °C, with humidity levels ranging from 30–70%.

**Flow cytometry.** For the analyses presented in Fig. 2i and Extended Data Fig. 10a,b, mouse HSCs and multipotent progenitors (MPPs) (cKit$^+$Lin$^-$Sca1$^+$, termed 'KLS'), common myeloid progenitors (CMPs) (cKit$^+$Lin$^-$Sca1$^{lo/-}$CD34$^{med/hi}$CD16/32$^{lo/-}$) and common lymphoid progenitors (CLPs) (cKit$^{lo}$Lin$^-$ Sca1$^{lo}$CD135$^+$ CD127$^+$) were isolated as described previously[59] (Extended Data Fig. 10a). In brief, hips, femurs, tibia and humeri were collected from C57BL/6 mice. Bones were cleaned, cut and flushed with a syringe filled with ice-cold FACS buffer (2% fetal bovine serum in Hanks' balanced salt solution buffer). Cells in FACS buffer were filtered through a 40-µm filter, pelleted and then incubated in ammonium–chloride–potassium (ACK) lysis buffer for 5 min on ice. Cells were then spun down and resuspended in 400 µl FACS buffer per mouse. Lineage depletion beads (Miltenyi Biotec 130-110-470) were added to the cells (50 µl per mouse) and incubated for 10 min at 4 °C. After incubation, the cells were loaded onto an LS magnetic separation column (Miltenyi Biotec 130-042-401), which was subsequently washed with 3 × 3 ml of FACS buffer. Before and after washing, pass-through cells were collected, spun down and resuspended in FACS buffer. For the isolation of KLS and CMP cells, the following antibodies were used: anti-mouse lineage cocktail-A700 (BioLegend 133313, 5 µl per mouse), anti-CD117 (cKit)-BV395 (Thermo Fisher Scientific 363-1171-80, 1:100 dilution), anti-Sca1-BV605 (BioLegend 108133, 1:100 dilution), anti-CD34-eFluor 450 (Thermo Fisher Scientific 48-0341-80, 1:40 dilution) and anti-CD16/32-BV711 (BD Biosciences 740659, 1:100 dilution). Following the addition of the anti-CD34 antibody, cells were incubated on ice for 45 min before adding the remaining antibodies. The cells were then incubated with the remaining antibodies for an additional 20 min on ice, followed by washing and FACS analysis. For the isolation of CLP cells, the following antibodies were used: anti-mouse lineage cocktail-A700 (BioLegend 133313, 5 µl per mouse), anti-CD117 (cKit)-BV395 (Thermo Fisher Scientific 363-1171-80, 1:100 dilution), anti-Sca1-BV605 (BioLegend 108133, 1:100 dilution), anti-CD135-BV421 (BioLegend 135313, 1:100 dilution) and anti-CD127 (IL-7Rα)-BV711 (BioLegend 135035, 1:100 dilution). The cells were incubated with the antibodies for 20 min followed by washing and FACS analysis. Flow cytometry was performed with a 100 µM nozzle on a BD FACSAria II using FACSDiva software (v.9.7).

Blood samples were collected from the same mice for the isolation of CD8a$^+$ T cells (CD3$^+$ CD8a$^+$) and B (CD19$^+$) cells. Peripheral blood mononuclear cell (PBMC) isolation was performed using a SepMate-15 tube (STEMCELL Technologies 85415) according to the manufacturer's instructions. Enriched PBMCs were resuspended in FACS buffer and incubated with either T cell antibodies (anti-CD3-BV711, BioLegend 100241, 1:100; anti-CD8a-BV605, BioLegend 100743, 1:100 dilution) or B cell antibodies (anti-CD19-BV605, BioLegend 115539, 1:100 dilution) on ice for 20 min. The cells were then washed with FACS buffer and analyzed on a BD FACSAria II using FACSDiva software (v.9.7). Flow cytometry data were analyzed with FlowJo (v.10.9.0).

**RNA isolation and real-time PCR.** For the analyses presented in Fig. 2i and Extended Data Fig. 10b, 20,000 sorted cells from each bone marrow and blood population noted in 'Flow cytometry' were lysed in RNA lysis buffer (RLT) and subjected to RNA extraction using the RNeasy Plus Micro kit (QIAGEN, 74034). RNA was then reverse transcribed into cDNA with SuperScript III First Strand Synthesis kit (Thermo Fisher Scientific, 11752-050) according to the manufacturer's instructions. Real-time quantitative PCR was conducted on the QuantStudio 7 PRO Real-Time PCR System utilizing Power SYBR Green PCR Master Mix (Thermo Fisher Scientific, 4368706). *Actb* was used as an internal control. The following qPCR primers were used (5' → 3'). *Actb*: Forward GATCATTGCTCCTCCTGAGC, Reverse ACTCCTGCTTGCTGATCCAC; *Hoxb5*: Forward CGATCCAC AAATCAAGCCC, Reverse TGCCACTGCCATAATTTAGC; *Fgd5*: Forward CTGGTTTTACTCCTGGTGAC, Reverse AGCTGATACTTCCTGTCT GG; *Procr*: Forward GGACTCGGTATGAACTGCA, Reverse CAGTGAT GTGTAAGAGCGAC; *Cd34*: Forward ACTATAAGCTTCCTCTCCTGG, Reverse ACACCCAATCCTCTCATCTC; *Cd8a*: Forward GAGAACATTC CTTAGCACCC, Reverse GCAGTTTTGACAGTCAGCG; *Cd19*: Forward AGGAAAAGGAAGCGAATGAC, Reverse GCCAGAGGTAGATGTAGGAAG; *Fads1*: Forward TGGTTTGGGAGGCATTTG, Reverse GCCATCCGTTTTG TCAAGAG; *Fads2*: Forward CAGGAGTGTAGAGGGAAGAG, Reverse CTCAGAATGACATAGCGTGG; *Scd2*: Forward ACTCTGCCTGGGATA CATG, Reverse CCCACCCCAAAACACAAAA.

**In situ hybridization and immunofluorescence.** Intestinal tissues analyzed in Fig. 2j and Extended Data Fig. 10c–e were collected from C57BL/6 mice, cleaned with cold PBS and fixed in 10% neutral buffered formalin at 4 °C overnight. Then, 7-µm optimal cutting temperature compound frozen sections were prepared for the RNAscope HiPlex12 Reagents Kit v.2 assay (Advanced Cell Diagnostics, 324409), which was performed according to the manufacturer's instructions with the following probes: Mm-*Lgr5*-T1 (Advanced Cell Diagnostics, 312171-T1), Mm-*Mki67*-T2 (Advanced Cell Diagnostics, 416771-T2), Mm-*Fads1*-T3 (Advanced Cell Diagnostics, 801641-T3), Mm-*Fads2*-T4 (Advanced Cell Diagnostics, 568621-T4), Mm-*Fgfbp1*-T5 (Advanced Cell Diagnostics, 508831-T5) and Mm-*Scd2*-T7 (Advanced Cell Diagnostics, 486111-T7). Protease Plus (Advanced Cell Diagnostics, 322331) was used for tissue pre-treatment. Following the last round of in situ hybridization imaging, fluorophores were cleaved using fresh 10% cleaving solution v.2. The intestinal tissues were then subjected to immunofluorescence staining. In brief, tissues were washed with PBS, permeabilized with 0.1% Triton X-100 in PBS and then blocked with 5% bovine serum albumin (BSA) in PBS for 30 min at room temperature. The tissues were then incubated with anti-E-Cadherin-Alexa Fluor 488 antibody (BD Biosciences 560061, 1:50 dilution) diluted in staining buffer (5% BSA in PBS with 0.1% Triton X-100) for 1 h at room temperature, followed by washing and imaging. All fluorescence images were acquired on a Zeiss LSM 980 confocal microscope. To quantify colocalization, cells along the crypts–villi axis were first categorized into different cell zones, as described in the caption of Extended Data Fig. 10e. Mean fluorescence intensities were then determined using ImageJ (v.1.53t).

### Statistics and reproducibility

Relationships between two ordered variables were assessed by correlation tests or linear regression. Unless otherwise stated, statistical significance for Kendall correlations was determined by a two-sided $z$-test. Two-group comparisons were assessed using unpaired or paired tests, as appropriate. Results with $P < 0.05$ were considered significant. No statistical method was used to predetermine sample size. Data analyses were performed using Python (v.3.9.0) and R (v.4.2.0+). Software packages and versions specific to each analysis are detailed in the Methods. For routine plotting and data manipulation, we also used the R packages ggplot2 (v.3.4.3), Matrix (v.1.6.1)

and dplyr (v.1.1.3), as well as the Python packages pandas (v.2.2.3) and numpy (v.1.26.3).

## Reporting summary

Further information on research design is available in the Nature Portfolio Reporting Summary linked to this article.

## Data availability

All datasets comprising the single-cell potency atlas assembled in this work (Supplementary Table 1) are publicly available from GEO, ArrayExpress or the SRA with the following accession codes: GSE52583 ('AT2/AT1 lineage (C1)'), GSE109774 ('Bone marrow (10x)', 'Bone marrow (Smart-seq2)' and 'Tabula Muris (Smart-seq2/10x)'), GSE60783 ('Dendritic cells (C1)'), GSE97391 ('Direct in vitro neuron (inDrop)' and 'Standard in vitro neuron (inDrop)'), GSE70245 ('HSPCs (C1)'), GSE113197 ('Human breast 1 (10x)' and 'Human breast 1 (C1)'), GSE161529 ('Human breast 2 (10x)'), GSE36552, ('Human embryo (Tang et al.) (ref. 60)') GSE92332 ('Intestine (Drop-seq)' and 'Intestine (Smart-seq2)'), GSE85066 ('Mesoderm (C1)'), GSE45719 ('Mouse embryo 1 (Tang et al.), (ref. 60)'), SRP073767 ('Peripheral blood (10x)'), GSE128639 ('BM-MNC (CITE-seq)'), GSE100866 ('Cord blood (CITE-seq)'), E-MTAB-9067 ('HSC development (Smart-seq2)'), GSE90742 ('HSCs and MPPs (inDrop)'), E-MTAB-11536 ('Immune cell atlas (10x)'), GSE76408 ('Lgr5-CreER intestine (CEL-seq)'), E-MTAB-3321 ('Mouse embryo 2 (Smart-seq2)'), GSE59892 ('Mouse embryo 3 (Smart-seq)'), GSE162044 ('Neural crest (Smart-seq2)'), GSE132188 ('Pancreas (10x)'), GSE99933 ('Peripheral glia (Smart-seq2)'), GSE122466 ('Retinal neurons (10x)'), GSE64447 ('Skeletal stem cell (C1)') and GSE201333 ('Tabula Sapiens (Smart-seq2/10x)').

Raw FASTQ or BAM files analyzed in this work are available from the SRA with the following accessions: SRP188993 ('BM-MNC (CITE-seq)'), SRP168426 ('Retinal neurons (10x)'), SRP200419 ('Pancreas (10x)'), SRP109011 ('Peripheral glia (Smart-seq2)'), SRP239468 ('Skeletal stem cell (C1)'), SRP094420 ('HSCs and MPPs (inDrop)') and SRP476153 ('Mouse neurogenesis (10x)' and 'Mouse mature neural cell types (10x)'). Five expression datasets covering mouse embryogenesis periods from implantation to organogenesis are accessible from GEO or ArrayExpress with the following accessions: GSE100597 ('Implantation (E3.5–E6.5)'), GSE109071 ('Implantation (E5.5–E6.5)'), E-MTAB-6967 ('Gastrulation (E6.5–E8.5)'), GSE186069 ('Organogenesis (E8.5)'), and GSE228590 ('Organogenesis (E8.75–P0)').

The publicly available oligodendroglioma and AML expression data analyzed in this work are available with GEO accession numbers GSE70630 and GSE116256, respectively.

Reference genomes and annotation files for GRCm39 (mouse) and GRCh38.p13 (human) were obtained from Ensembl release 109 (February 2023) via the archive at https://feb2023.archive.ensembl.org.

## Code availability

R and Python packages for running CytoTRACE 2 with the pre-trained model are freely available for non-profit academic use at https://github.com/digitalcytometry/cytotrace2. Both packages implement optional parallel processing for efficient execution and provide built-in plotting functions (UMAPs and box plots). Documentation, vignettes and input examples are provided. The Python version of the package is also available via PyPI at https://pypi.org/project/cytotrace2-py/. An interactive RShiny web application is available, allowing users to:

- Run CytoTRACE 2 on user-provided datasets via an intuitive, interactive interface
- Browse CytoTRACE 2 results for 33 datasets with ground truth potency annotations
- Explore potency-associated genes learned by CytoTRACE 2 and investigate potency enrichment of user-defined genes and gene sets across the single-cell potency atlas

- Download the single-cell potency atlas
- Access tutorials, vignettes and FAQs, along with additional Python code and vignettes for training the CytoTRACE 2 model and for creating custom GSBN architectures.

This application can be accessed at https://cytotrace2.stanford.edu.

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

## Acknowledgements

The authors acknowledge A. Gentles, S. Sikandar, A. Usmani, N. Earland, R. Fields and D. Chen for providing critical feedback on this manuscript. We are grateful to M. Haney for sharing CRISPR screening data, A. Alizadeh and C.L. Liu for providing RNAscope equipment and S. Thapa for initial software testing. This work was supported by a Stanford Bio-X Interdisciplinary Graduate Fellowship (M.K.), a Stanford Dean's Postdoctoral Fellowship (J.J.A.A.), the National Science Foundation (R.G., J.P.D. and J.S., Graduate Research Fellowship DGE-1656518), the Norwegian Research Council (C.B.S., 334328), Stanford Graduate Fellowship in Science & Engineering (J.S.), the V Foundation for Cancer Research V Scholar Award (A.A.C.), the National Cancer Institute (A.A.C. and A.M.N., R01CA283317; A.M.N., R01CA255450), the Virginia and D. K. Ludwig Fund for Cancer Research (A.M.N.) and the Donald E. and Delia B. Baxter Foundation (A.M.N.). A.M.N. is a Chan Zuckerberg Biohub – San Francisco Investigator.

## Author contributions

A.M.N., M.K., G.S.G. and E.L.B. conceived of the study, developed strategies for related experiments and wrote the paper with substantial assistance from Z.Q. J.J.A.A. assisted with the initial development of CytoTRACE 2. M.K., E.L.B., J.J.A.A. and A.M.N. implemented CytoTRACE 2 with assistance from S.A. G.S.G. determined potency annotations with assistance from M.K. and A.M.N. S.A., R.G., J.S., C.B.S., J.P.D., W.Z. and A.A.C. assisted with data analysis and interpretation. Z.Q. performed experimental validation experiments under the supervision of A.M.N. and M.F.C. All authors commented on the paper at all stages.

## Competing interests

M.K., G.S.G., E.B., J.J.A.A. and A.M.N. have a patent application (US PCT/US25/18429) related to the identification of developmental states from scRNA-seq data. A.M.N. has ownership interests in CiberMed and LiquidCell Dx. A.A.C. has ownership interests in Droplet Biosciences and LiquidCell Dx. All other authors declare no competing interests.

## Additional information

**Extended data** is available for this paper at https://doi.org/10.1038/s41592-025-02857-2.

**Correspondence and requests for materials** should be addressed to Aaron M. Newman.

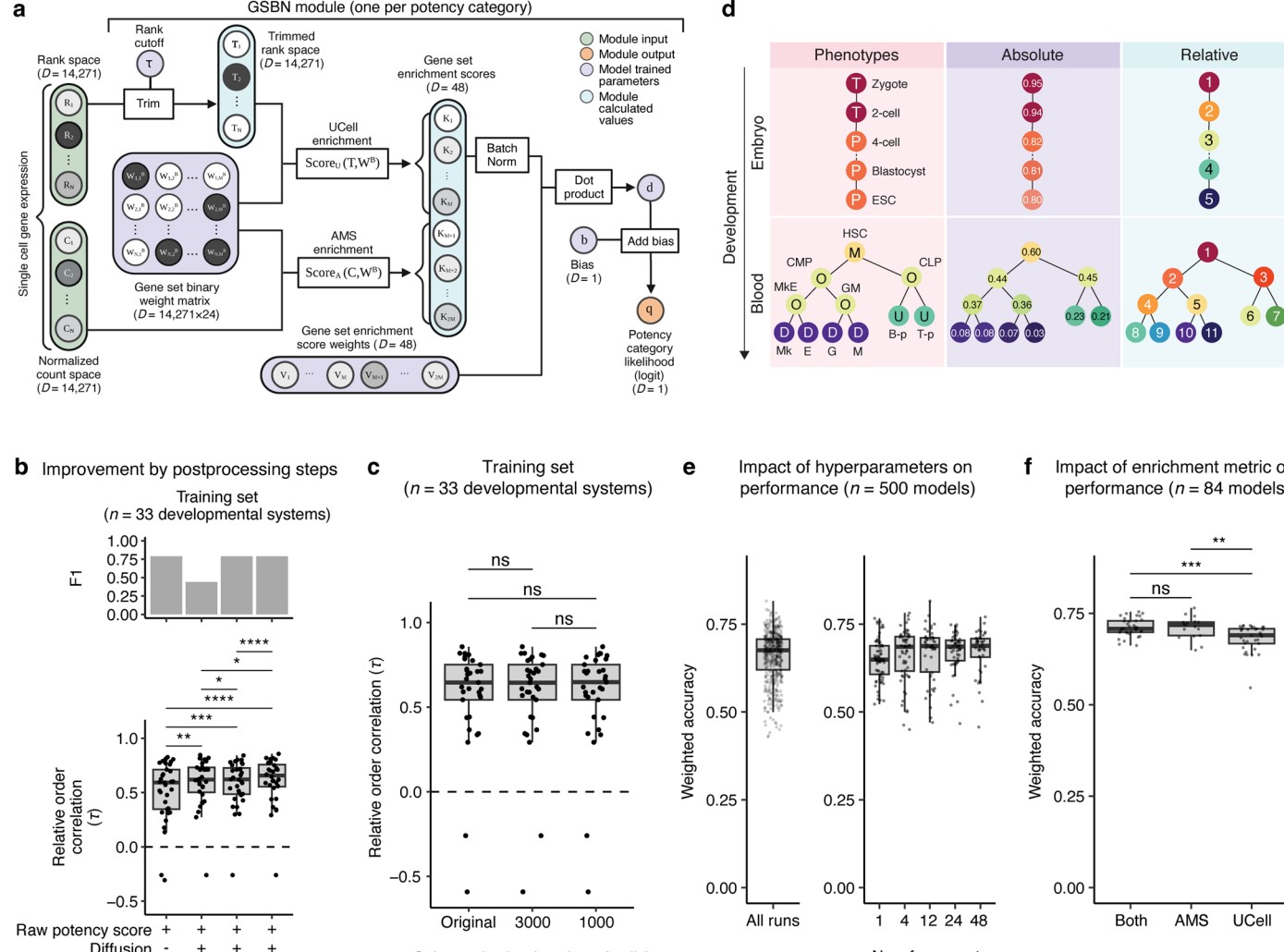

**Extended Data Fig. 1 | CytoTRACE 2 architecture and benchmarking metrics. a**, Schematic overview of the CytoTRACE 2 core model, focusing on architectural details and key operations between the input layer and output layer for a single gene set binary network (GSBN) module. Note that six GSBN modules, one per broad potency category, are included in the full model, as illustrated in Fig. 1c. For additional details, see Methods. **b**, Serial impact of three postprocessing procedures ("*Postprocessing*" in Methods) on the accuracy of predicting (i) relative developmental orderings (n = 33 systems; bottom) and (ii) potency classes (F1 score, mean-aggregated across six broad potency categories; top) on training datasets (Methods). All results were obtained via leave-one-out cross-validation (LOOCV). **c**, Box plot showing the performance of CytoTRACE 2 for the prediction of relative orderings (Supplementary Table 4) after subsampling cells for the Markov diffusion step of the postprocessing procedure (Methods). All training datasets were analyzed using LOOCV (Supplementary Table 1). **d**, Illustration of the difference between absolute and relative developmental potential using two hypothetical scRNA-seq datasets, one spanning totipotent through pluripotent cells ('Embryo') and the other encompassing multipotent through differentiated cells ('Blood'). Hypothetical prediction scores are shown for absolute and relative orderings, with the latter reset for each dataset (for example, as for CytoTRACE 1, RNA velocity, Monocle, and CellRank). ESC, embryonic stem cell; HSC, hematopoietic stem

cell; CMP, common myeloid progenitor; CLP, common lymphoid progenitor; MkE, megakaryocyte-erythroid progenitor; GM, granulocyte progenitor; B-p, B cell progenitor; T-p, T cell progenitor; Mk, megakaryocyte; E, erythrocyte; G, granulocyte; M, monocyte. **e-f**, Impact of hyperparameter values on training set performance, showing weighted accuracy for single-cell potency classification using nested LOOCV (Supplementary Table 6; Methods). **e**, *Left*: Each point (n = 500) denotes the results from one iteration of the hyperparameter sweep. *Right:* Same as the left but showing weighted accuracy plotted by the number of gene sets per GSBN module. **f**, Weighted accuracy of models trained using distinct gene set enrichment procedures (AMS and/or UCell) after selecting robust hyperparameter values as described in Methods. Statistical significance was determined using a two-sided paired Wilcoxon test in panels **b** and **c** and a two-sided unpaired Wilcoxon test in panel **f**. *P < 0.05; **P < 0.01; ***P < 0.001; ****P < 0.0001; ns, not significant. Note that tissue-specific expression matrices from plate- and droplet-based platforms within Tabula Muris[43] were analyzed individually in panels **b** and **c** for clarity, yielding 33 total systems with known developmental orderings. In **b**, **c**, **e**, and **f**, the box center lines, bounds of the box, and whiskers denote medians, 1st and 3rd quartiles, and minimum and maximum values within 1.5 × IQR (interquartile range) of the box limits, respectively. Panel **a** was created using BioRender.com.

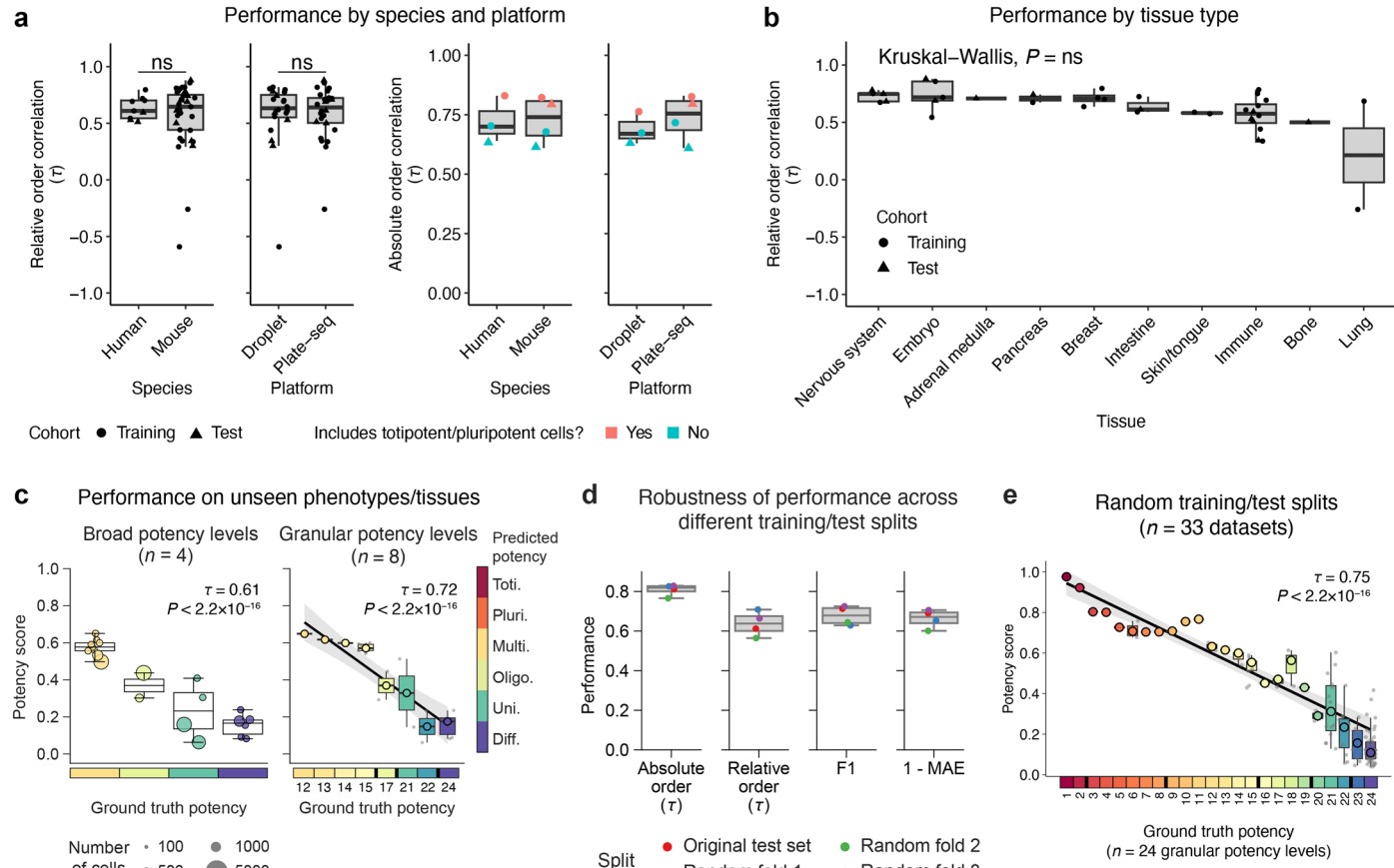

**Extended Data Fig. 2 | Validation and generalizability of CytoTRACE 2.**
**a**, Performance of CytoTRACE 2 for reconstructing relative developmental trajectories (left) and six broad potency categories (right) in 33 ground truth scRNA-seq datasets (Fig. 1b), stratified by species and platform. Performance was evaluated at the single-cell level using weighted Kendall correlation (τ), as described in Supplementary Table 5 and Methods. To promote a fair comparison, we evaluated absolute order correlations (right) with and without the inclusion of totipotent and pluripotent cells, as the corresponding potency categories were not available for human and droplet-only datasets in the test cohort. ns, not significant (two-sided unpaired Wilcoxon test). **b**, Same as panel **a** (left) but showing relative order performance stratified by developmental clade (Fig. 1b). ns, not significant (Kruskal-Wallis test). **c**, Same as Fig. 1d-e but showing performance on 21 cell phenotypes that were unseen during model training, including cranial neural crest cells, apical progenitors, skeletal stem cells,

epsilon cells, and photoreceptor cells (Supplementary Table 7). **d**, Performance of CytoTRACE 2 on different training-test configurations, comparing the original split ('Original test set'; Fig. 1b) with three additional splits, the latter of which were randomized to balance potency categories between cohorts (Supplementary Table 8; Methods). Performance was evaluated with four metrics, each calculated at the single-cell level in held-out data: absolute order (weighted τ), relative order (median weighted τ across evaluable systems with known developmental orderings), multiclass F1 for predicting broad potency classes (n = 6), and one minus the mean absolute error (MAE) for predicting broad potency classes (n = 6). For details, see Methods. **e**, Same as Fig. 1e but showing held-out data from three random training/test splits in **d**. In **a-e**, the box center lines (a-d) and circles (c right and e), bounds of the box, and whiskers denote medians, 1st and 3rd quartiles, and minimum and maximum values within 1.5 × IQR (interquartile range) of the box limits, respectively.

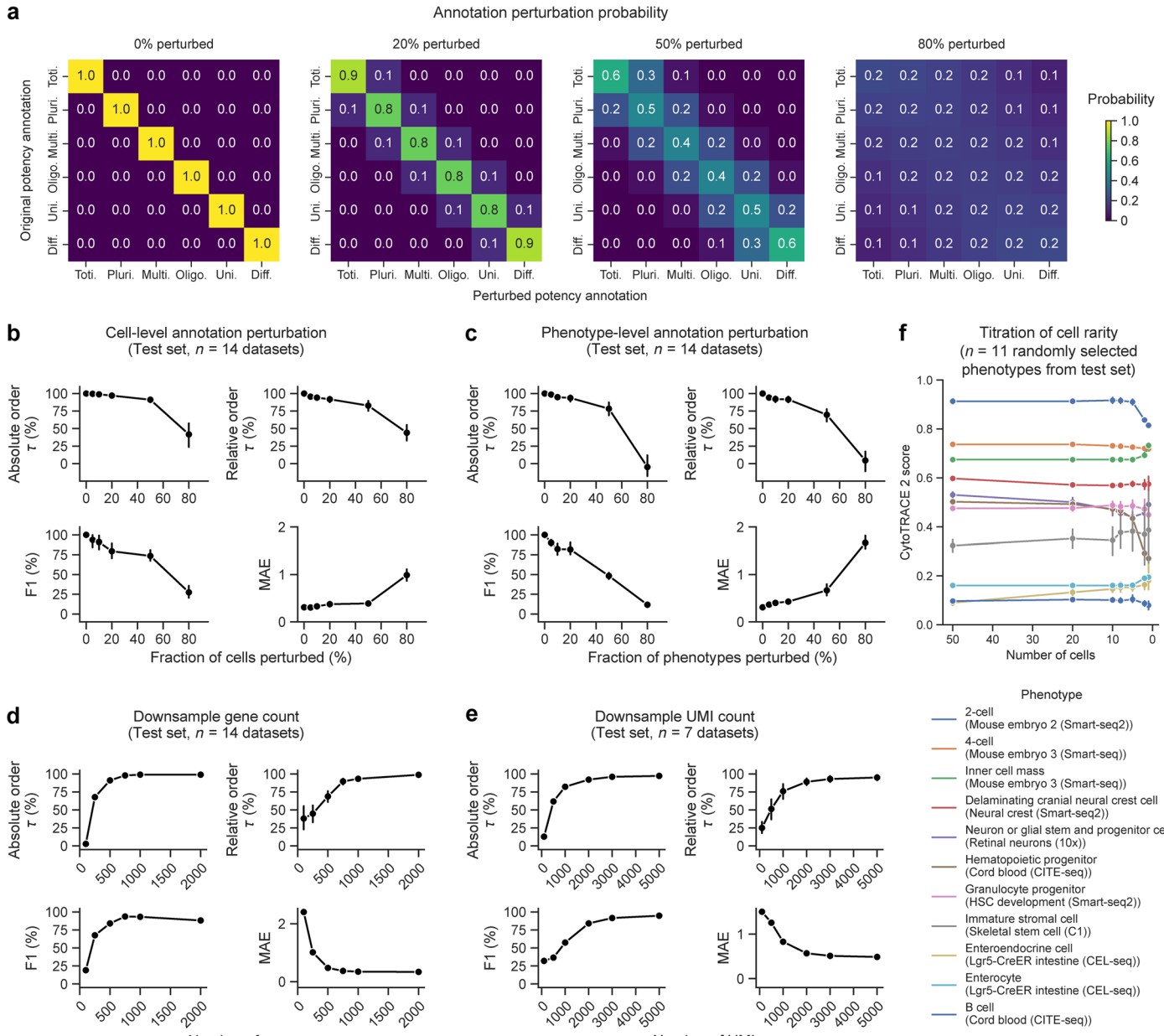

**Extended Data Fig. 3 | Robustness of CytoTRACE 2. a-c,** Impact of perturbing potency labels in the training set. **a,** Heat maps depicting individual potency labels and their transition probabilities at different perturbation levels (Methods). **b-c,** Performance of CytoTRACE 2 models trained on potency labels with defined perturbation levels (as illustrated in **a** and described in Methods) applied to individual cells (**b**) or phenotypes (**c**) in the training set. Performance in held-out test data was evaluated with four metrics: absolute order (weighted τ of broad potency levels across datasets), relative order (median weighted τ of each dataset analyzed individually), multiclass F1 for predicting broad potency classes ($n = 6$), and the mean absolute error (MAE) for predicting broad potency classes ($n = 6$). For details, see Methods. Absolute order (τ), relative order (τ), and F1 score are expressed as a percentage of the results obtained with the unperturbed CytoTRACE 2 model. Each point represents the mean across

five replicates of random perturbation. Error bars represent 95% confidence intervals. **d,** Analysis of robustness to the number of genes per cell using all test datasets ($n = 14$) (Supplementary Table 1) assessed with the same metrics as panels **b** and **c**. Results for each dataset represent the average across five rounds of gene count downsampling and are expressed relative to results with no downsampling. **e,** Same as panel **d** but shown for UMIs per cell in evaluable test datasets ($n = 7$). **f,** Impact of the number of cells per phenotype on the consistency of CytoTRACE 2 potency scores in test datasets. Eleven phenotypes spanning a range of potencies were titrated in defined amounts ($x$-axis) while other cells were left unchanged. CytoTRACE 2 was then applied to predict potency scores. Points represent averages from five random samplings (without replacement) per phenotype and error bars represent 95% confidence intervals.

**Extended Data Fig. 4 | CytoTRACE 2 correctly identifies a transient pluripotent cell state during mouse cranial neural crest development.** Same as Fig. 1g but focusing on mouse cranial neural crest cells profiled by Zalc et al.[11]. *Left inset:* Log expression levels of core pluripotency factor *Pou5f1* in mouse cranial neural crest cells (CNCCs). *Right inset:* Cells predicted as pluripotent by CytoTRACE 2. Cells predicted as pluripotent by CytoTRACE 2 showed significantly higher *Pou5f1* expression than others, with $P = 2.6 \times 10^{-5}$ by two-sided Wilcoxon test.

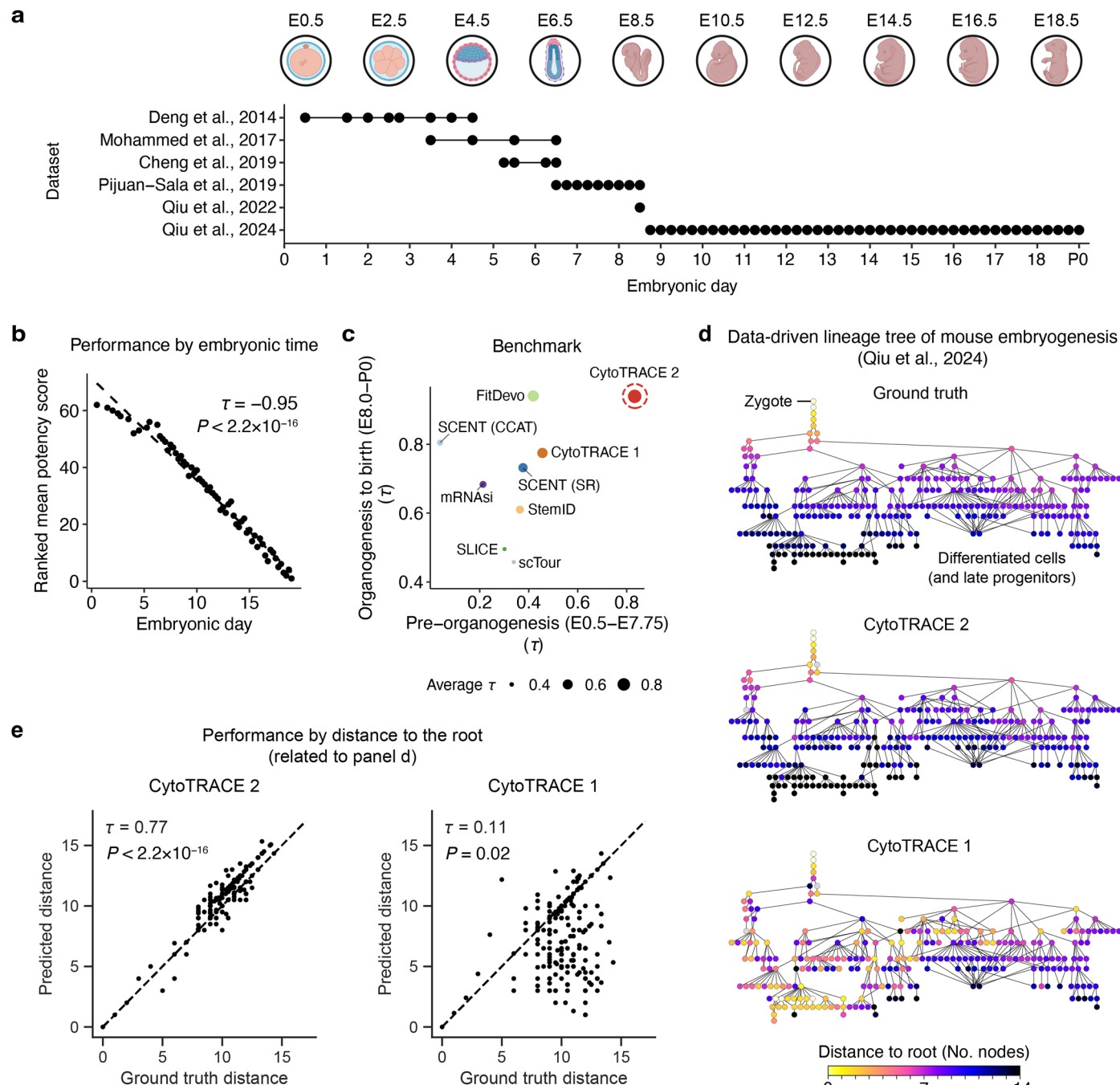

**Extended Data Fig. 5 | Large-scale reconstruction of cell potency during mouse embryogenesis. a**, Overview of single-cell expression datasets analyzed in **b**-**e** and corresponding developmental time points profiled (*n* = 62). Icons denoting key stages of mouse embryogenesis were created using BioRender.com. **b**, Linearity between the average CytoTRACE 2 potency score per time point (weighted equally across author-annotated phenotypes) expressed in rank space (*y*-axis) and the corresponding time points (*n* = 62; *x*-axis). Concordance was calculated using linear regression (dashed line) and Kendall correlation (τ), with the latter weighted by the number of time points per embryonic day. **c**, Scatter plot comparing the performance of CytoTRACE 2 to previous approaches for reconstructing the temporal hierarchy of 45 time points spanning organogenesis (beginning at E8.0[61]) to birth (*y*-axis) versus 17 time points preceding organogenesis (*x*-axis). Correlations are weighted by whole day intervals to account for imbalances in the number of evaluable time points per day. Point sizes represent the average weighted Kendall correlation per approach. **d**, *Top:* Data-driven lineage tree of mouse embryogenesis, where nodes represent cell types (*n* = 259), edges represent developmental transitions inferred by Qiu et al.[2], and colors represent the corresponding rank distance from each cell type to the root ("Ground truth"). *Center and bottom*: Same as top, but with CytoTRACE 2 and CytoTRACE 1 predictions each averaged by phenotype, then rank-ordered along the path to the root. Lower ranks indicate shorter distances. Distances were averaged for cell types with multiple direct paths to the root. **e**, Scatter plots showing all distances in **d**, with concordance between CytoTRACE methods (center and bottom panels of **d**) and lineages inferred by Qiu et al.[2] (top panel of **d**). The significance of τ in **b** and **e** was determined using a two-sided Z-test.

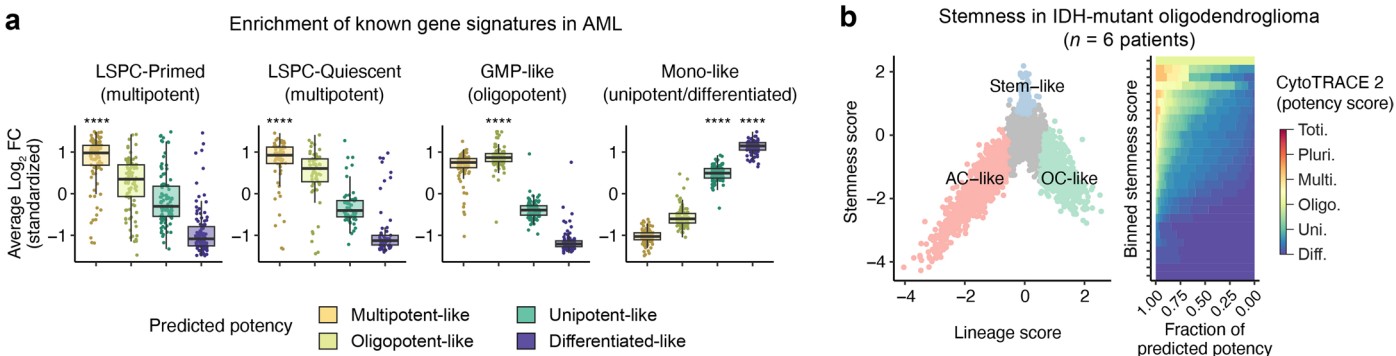

**Extended Data Fig. 6 | Tracing developmental lineages in AML and oligodendroglioma. a**, Box plots showing relative expression levels of cell state signatures from patients with acute myeloid leukemia (AML)[26] in 13,445 AML cells stratified by potency categories identified by CytoTRACE 2. Each point denotes a single gene from the corresponding gene set ID indicated above the plot (Supplementary Table 10). Genes were internally normalized within each tumor sample as the mean log$_2$ fold change (FC) within a given potency category versus the remaining cells in the tumor, as described in Methods, then z-score normalized (standardized) across potency categories. The four signatures, LSPC-Primed, LSPC-Quiescent, GMP-like, and Mono-like, are expected to be most highly expressed in multipotent, multipotent, oligopotent, and unipotent/differentiated cells, respectively (Supplementary Table 10). Statistical significance comparing the expected potency level(s) with each other potency level was determined by a two-sided Wilcoxon test. ****$P$ < 0.0001. Box center lines, bounds of the box, and whiskers denote medians, 1st and 3rd quartiles, and minimum and maximum values within 1.5 × IQR (interquartile range) of the box limits, respectively. **b**, *Left:* Scatter plot of oligodendroglioma cells from six tumors organized by previously described stemness and lineage enrichment scores[27]. *Right:* Stacked bar plot showing how the fractional representation of cells with predicted potency categories (CytoTRACE 2) changes as a function of author-supplied stemness scores (*y*-axis). Cells predicted to have the highest oligo- and multilineage potential by CytoTRACE 2 correspond to those annotated as stem-like by Tirosh et al.[27]. Potency colors reflect eight evenly spaced bins per potency category.

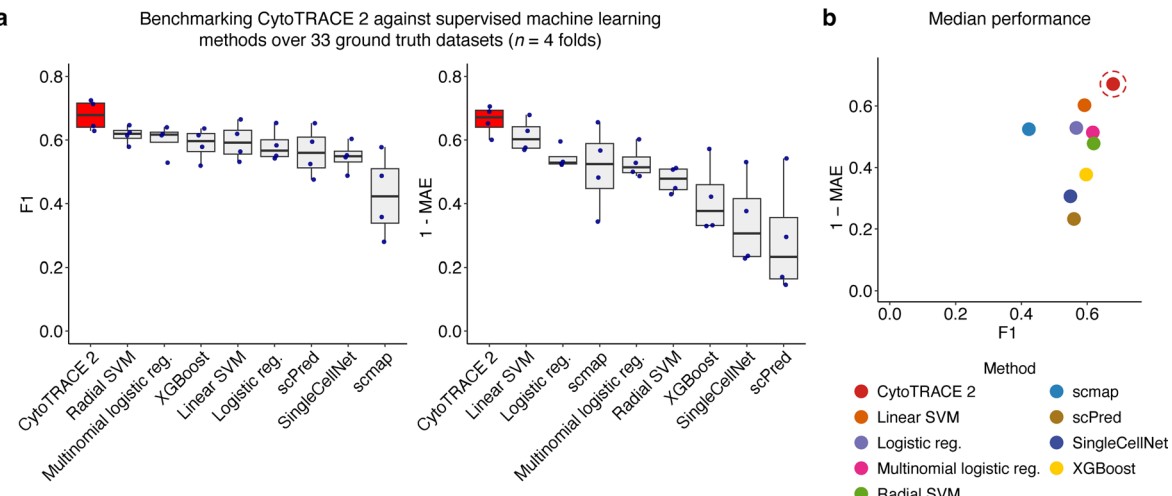

**Extended Data Fig. 7 | Benchmarking against supervised machine learning methods. a**, Box plots comparing the performance of CytoTRACE 2 against eight baseline methods (supervised machine learning models, including leading tools for reference-guided annotation of scRNA-seq data) implemented for single-cell potency classification (Methods). Each method was trained to assign cells to six broad potency categories using identical training-test splits. Four-fold cross-validation was performed for each method, where each point represents performance in one fold of held-out data (the original training-test split [Fig. 1b] and three random splits [Supplementary Table 8]). Performance was assessed at the single-cell level using multiclass F1 (left) and one minus the mean absolute error (MAE; right) for predicting broad potency classes (n = 6). Box center lines, bounds of the box, and whiskers denote medians, 1st and 3rd quartiles, and minimum and maximum values within 1.5 × IQR (interquartile range) of the box limits, respectively. **b**, Scatter plot comparing median performance scores for all methods from panel **a**.

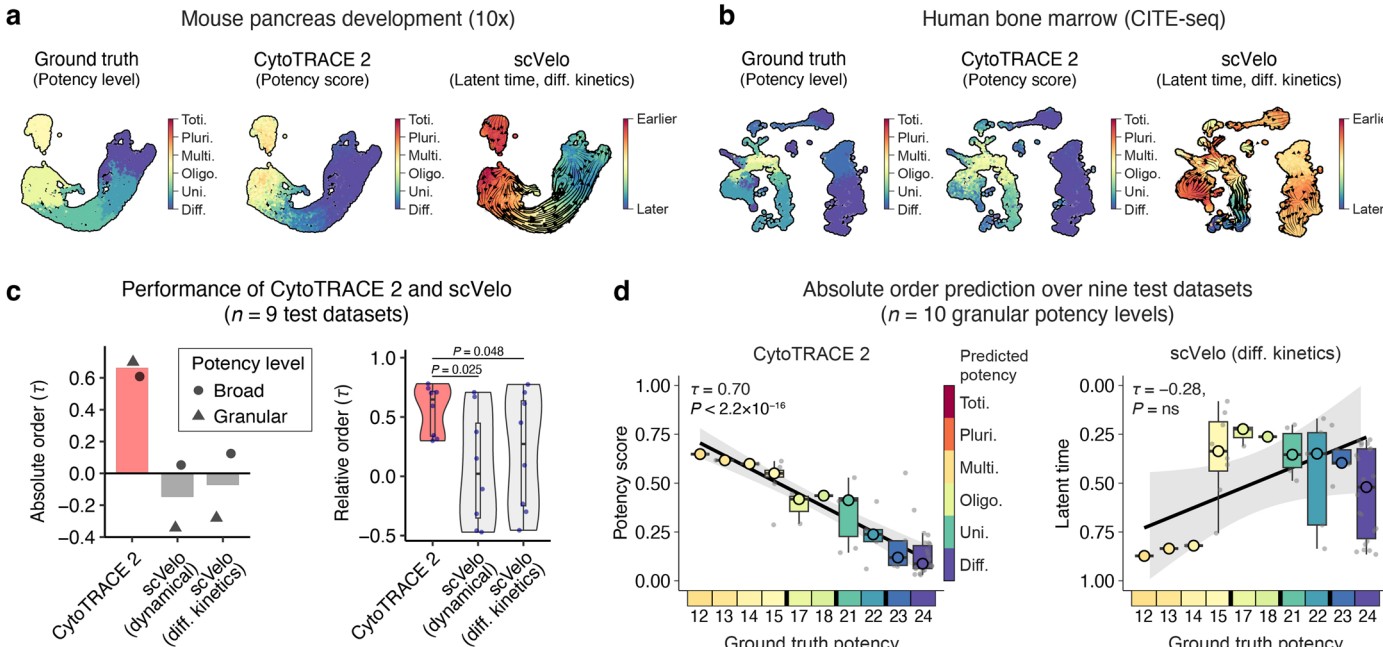

**Extended Data Fig. 8 | Benchmarking against scVelo. a-b**, Representative test datasets comparing CytoTRACE 2 and scVelo. **a**, UMAP representation of mouse pancreas development (10x) (Supplementary Table 1). *Left*: Cells colored by ground truth granular potency level (Fig. 1b; Supplementary Table 3). *Center*: Cells colored by CytoTRACE 2 potency scores. *Right*: Cells colored by scVelo latent time (differential kinetics model). **b**, Same as **a** but showing human bone marrow (CITE-seq) (Supplementary Table 1). **c**, *Left*: Bar plot showing mean absolute order (weighted τ applied to single cells) performance across six broad potency levels (circles) and ten granular order potency levels (triangles) for nine test datasets evaluable by CytoTRACE 2 and scVelo (Supplementary Tables 3 and 14; Methods). Two models are shown for the latter: dynamical latent time

and differential kinetics latent time. *Right*: Violin and box plots showing relative order performance (weighted τ applied to single cells) on the same test datasets (*n* = 8 evaluable datasets with relative developmental orderings, Supplementary Tables 4 and 14). Statistical significance was determined by two-sided paired *t* test. Violin plot bounds denote minimum and maximum values. Box center lines, bounds of the box, and whiskers denote medians, 1st and 3rd quartiles, and minimum and maximum values within 1.5 × IQR (interquartile range) of the box limits, respectively. **d**, Same as Fig. 1e but shown for the nine evaluable test datasets in **c**. *Left*: CytoTRACE 2 potency scores. *Right*: scVelo latent time (differential kinetics model). Statistical significance was determined using a one-sided Z-test. ns, not significant.

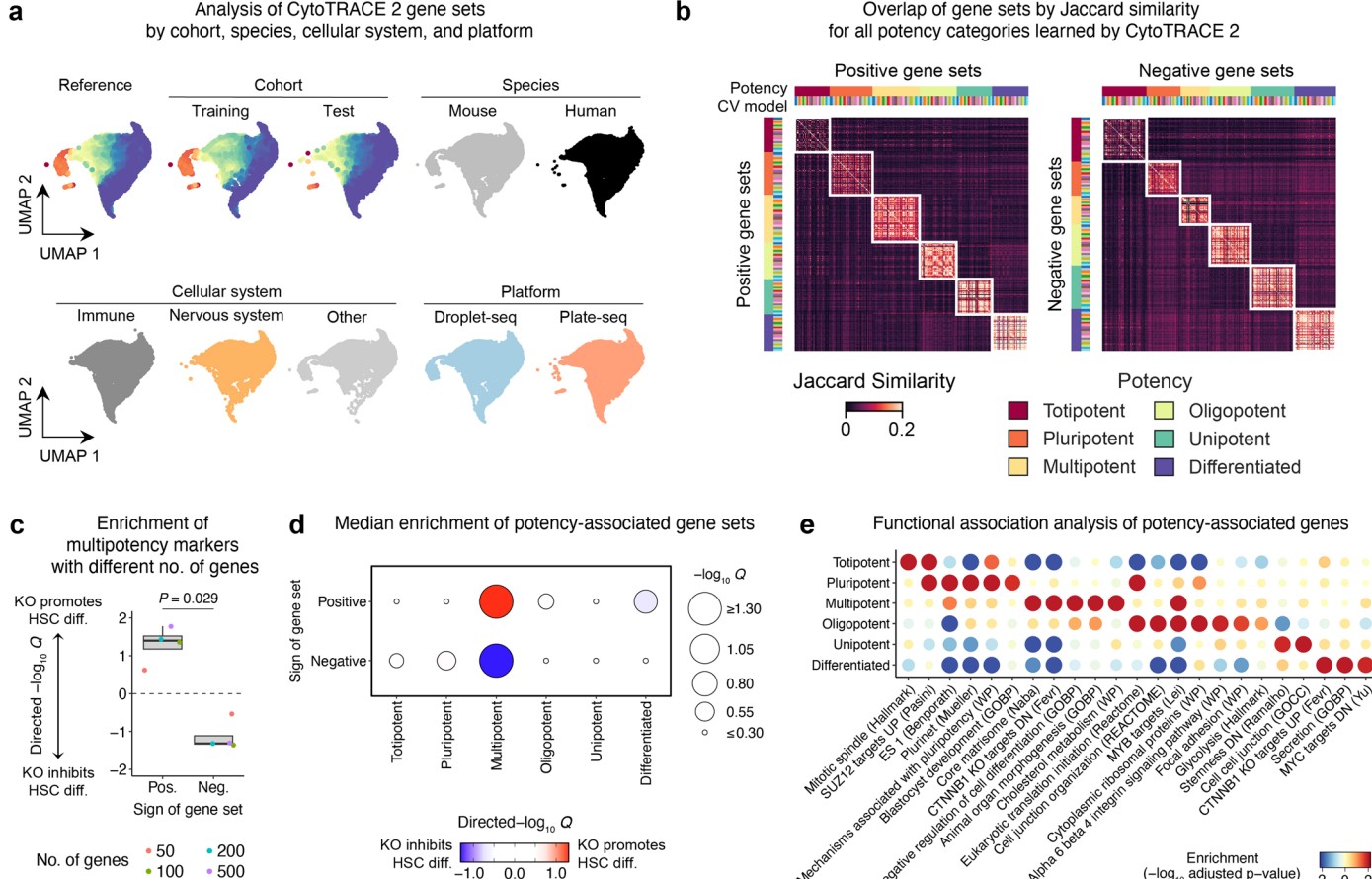

**Extended Data Fig. 9 | Extended analysis of potency programs and genes.**
**a**, Same as Fig. 2b but separated by cohort, species, cellular system (three general categories shown for clarity), and scRNA-seq platform. The embedding in Fig. 2b is shown as a reference in the upper left. Colors denote potency scores (same as the color bar in Fig. 2b, top) for reference and cohort-stratified embeddings. **b**, Heat map depicting pairwise similarity of gene sets learned by CytoTRACE 2 across all 19 ensemble models from leave-one-out cross-validation on the 19-dataset training cohort. Overlap was quantified by Jaccard index and stratified into gene sets with positive (left, $n = 1,490$) and negative weights (right, $n = 1,246$); gene set polarity was determined as described in "*Interpretability*," Methods. **c**, Same as Fig. 2e but showing the consistency between CytoTRACE 2 multipotency markers and hematopoietic stem cell (HSC) knockout (KO) phenotypes across a range of top $k$ markers, whether positive or negatively associated with multipotency ($k = 50, 100, 200,$ and $500$). GSEA statistics are expressed as directed $-\log_{10} Q$ values. Statistical significance between groups

was determined using a two-sided unpaired Wilcoxon test. Box center lines, bounds of the box, and whiskers denote medians, 1st and 3rd quartiles, and minimum and maximum values within $1.5 \times$ IQR (interquartile range) of the box limits, respectively. **d**, Same as **c**, but showing the median directed $-\log_{10} Q$ value across all top $k$ markers shown in **c**, stratified by positive and negative markers, and extended to all potency categories in the CytoTRACE 2 feature importance matrix (Supplementary Table 15). **e**, Enrichments of selected gene sets from MSigDb in the CytoTRACE 2 feature importance matrix (Fig. 2a, right; Supplementary Table 15). Bubbles are colored by signed $-\log_{10}$ adjusted p-values (adjusted for multiple comparisons) calculated by GSEA, where the sign is determined by the direction of association between the genes and the potency category. All $-\log_{10}$ adjusted p-values, including those exceeding the color bar range, are provided in Supplementary Table 17. Bubble sizes are proportional to unsigned $-\log_{10}$ adjusted p-values within the color bar.

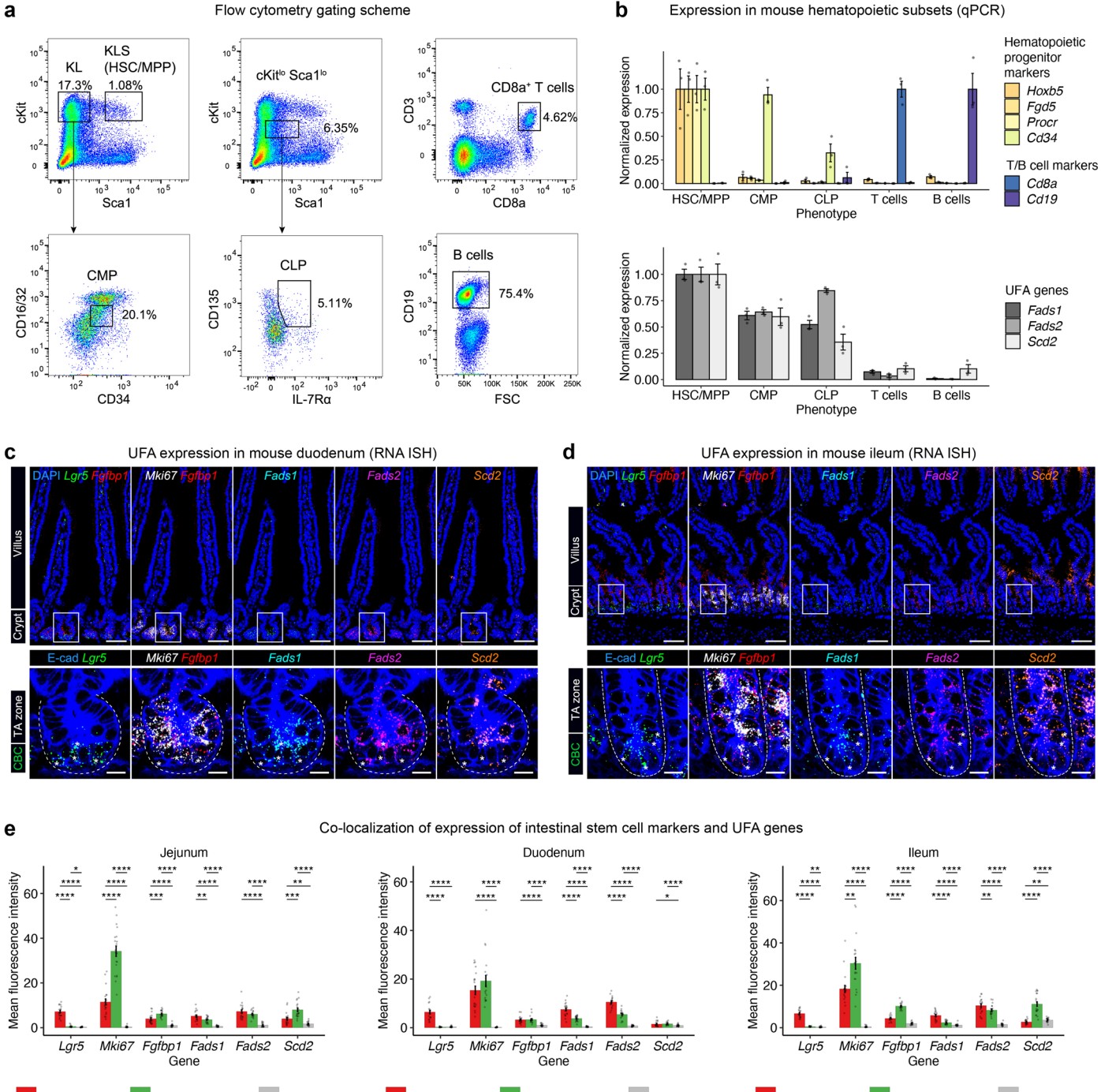

**Extended Data Fig. 10 | Validation of multipotency-associated genes.**
**a**, Representative gating schemes for FACS-purification of mouse hematopoietic subsets analyzed in Fig. 2i (HSCs/MPPs, CMPs, and CLPs from bone marrow and T/B cells from peripheral blood). KLS, Kit⁺ Lin⁻ Sca1⁺ multipotent cell subset consisting of HSCs and MPPs (multipotent progenitors); KL, Kit⁺ Lin⁻ Sca1⁻ subset devoid of multipotent cells; CMP, common myeloid progenitor; CLP, common lymphoid progenitor. **b**, Bar plots showing biological replicates and controls for quantitative PCR experiments, related to Fig. 2i. Each gene is shown normalized to the maximum mean expression across all groups. *Top*: markers of HSC/MPP (*Hoxb5*, *Fgd5*, *Procr*), progenitors (*Cd34*), and differentiated lineages (*Cd8a*, *Cd19*). *Bottom*: unsaturated fatty acid (UFA) synthesis genes identified as markers of multipotency by CytoTRACE 2 (*Fads1*, *Fads2*, and *Scd2*). *Actb* was used as an internal control. Error bars reflect s.e.m. (standard error of the mean) across three biological replicates. **c-d**, Same as Fig. 2j but shown for mouse duodenum

(**c**) and ileum (**d**). Scale bars, 50 μm (top) and 10 μm (bottom). **e**, Quantification of mRNA hybridization signal in multipotent and unipotent/differentiated zones of mouse jejunum (left, corresponding to images in Fig. 2j), duodenum (center, corresponding to confocal images in **c**), and ileum (right, corresponding to images in **d**). Multipotent zones are divided as previously described[40] on the basis of cell location from the crypt base, with red and green regions expected to be most enriched in *Lgr5* and *Fgfbp1*, respectively. Bars represent the mean fluorescence intensity per zone, with error bars denoting s.e.m. (*n* = 20 paired crypts and villi from each intestinal region [jejunum, duodenum, ileum] pooled from two mice, with a total of 10 paired crypts and villi per region per mouse). Statistical significance was determined by a two-sided paired *t* test, with the resulting p-values adjusted by the Benjamini-Hochberg method separately applied to jejunum, duodenum, and ileum samples (*Q < 0.05; **Q < 0.01, ***Q < 0.001, ****Q < 0.0001).

# Reporting Summary

## Statistics

For all statistical analyses, confirm that the following items are present in the figure legend, table legend, main text, or Methods section.

| n/a | Confirmed | |
|---|---|---|
| ☐ | ☒ | The exact sample size (*n*) for each experimental group/condition, given as a discrete number and unit of measurement |
| ☐ | ☒ | A statement on whether measurements were taken from distinct samples or whether the same sample was measured repeatedly |
| ☐ | ☒ | The statistical test(s) used AND whether they are one- or two-sided *Only common tests should be described solely by name; describe more complex techniques in the Methods section.* |
| ☐ | ☒ | A description of all covariates tested |
| ☐ | ☒ | A description of any assumptions or corrections, such as tests of normality and adjustment for multiple comparisons |
| ☐ | ☒ | A full description of the statistical parameters including central tendency (e.g. means) or other basic estimates (e.g. regression coefficient) AND variation (e.g. standard deviation) or associated estimates of uncertainty (e.g. confidence intervals) |
| ☐ | ☒ | For null hypothesis testing, the test statistic (e.g. *F*, *t*, *r*) with confidence intervals, effect sizes, degrees of freedom and *P* value noted *Give P values as exact values whenever suitable.* |
| ☒ | ☐ | For Bayesian analysis, information on the choice of priors and Markov chain Monte Carlo settings |
| ☐ | ☒ | For hierarchical and complex designs, identification of the appropriate level for tests and full reporting of outcomes |
| ☐ | ☒ | Estimates of effect sizes (e.g. Cohen's *d*, Pearson's *r*), indicating how they were calculated |

*Our web collection on statistics for biologists contains articles on many of the points above.*

## Software and code

Policy information about availability of computer code

| Data collection | Standard FACSDiva software (v9.7) was used for flow cytometry on a BD FACSAria II. Fluorescence images were acquired on a Zeiss LSM 980 confocal microscope. Publicly available FASTQ files were downloaded using sra-tools v3.1.1. |
|---|---|
| Data analysis | Software packages used in this study are detailed in Methods, including CytoTRACE 1 v0.3.3, scPred v1.9.2, pySingleCellNet v0.1.1, scmap v1.26.0, scikit-learn v1.1.1 & v1.4.2, XGBoost v2.1.1, SCENT v1.0.3, FitDevo v1.2.0, SLICE v0.99.0, RaceID v0.1.4, scTour v1.0.0, org.Hs.eg.db v3.15.0, cutadapt v4.9, dropest v0.8.6, STAR v2.7.11b, Cell Ranger v8.0.1, velocyto.py v0.17.17, scVelo v0.3.1, and wandb v0.16.4. Seurat versions 4.3.0 and 5.1.0, fgsea v1.25.1, GSVA v1.46.0, RANN v2.6.1, HiClimR v2.2.1, and various R v4.2+ packages (e.g., ggplot2 v3.4.3, matrix v1.6.1, dplyr 1.1.3) and python v3.9+ packages (e.g., pandas v2.2.3, numpy v1.26.3) were also used. CytoTRACE 2 results were generated with version 1.1.0.3 (cytotrace2-py) which uses python v3.9.0 and PyTorch v2.0.0.<br><br>Flow cytometry data was analyzed with FlowJo (v10.9.0). Fluorescence images were analyzed with ImageJ (v1.53t) to obtain mean fluorescence intensities. |

For manuscripts utilizing custom algorithms or software that are central to the research but not yet described in published literature, software must be made available to editors and reviewers. We strongly encourage code deposition in a community repository (e.g. GitHub). See the Nature Portfolio guidelines for submitting code & software for further information.

## Data

Policy information about availability of data

All manuscripts must include a data availability statement. This statement should provide the following information, where applicable:
- Accession codes, unique identifiers, or web links for publicly available datasets
- A description of any restrictions on data availability
- For clinical datasets or third party data, please ensure that the statement adheres to our policy

All datasets comprising the single-cell potency atlas assembled in this work (Supplementary Table 1) are publicly available from the Gene Expression Omnibus (GEO), ArrayExpress, or the Sequence Reach Archive (SRA) with the following accessions: GSE52583 ('AT2/AT1 lineage (C1)'), GSE109774 ('Bone marrow (10x)', 'Bone marrow (Smart-seq2)', and 'Tabula Muris (Smart-seq2/10x)'), GSE60783 ('Dendritic cells (C1)'), GSE97391 ('Direct in vitro neuron (inDrop)' and 'Standard in vitro neuron (inDrop)'), GSE70245 ('HSPCs (C1)'), GSE113197 ('Human breast 1 (10x)' and 'Human breast 1 (C1)'), GSE161529 ('Human breast 2 (10x)'), GSE36552 ('Human embryo (Tang et al.)'), GSE92332 ('Intestine (Drop-seq)' and 'Intestine (Smart-seq2)'), GSE85066 ('Mesoderm (C1)'), GSE45719 ('Mouse embryo 1 (Tang et al.)'), SRP073767 ('Peripheral blood (10x)'), GSE128639 ('BM-MNC (CITE-seq)'), GSE100866 ('Cord blood (CITE-seq)'), E-MTAB-9067 ('HSC development (Smart-seq2)'), GSE90742 ('HSCs and MPPs (inDrop)'), E-MTAB-11536 ('Immune cell atlas (10x)'), GSE76408 ('Lgr5-CreER intestine (CEL-seq)'), E-MTAB-3321 ('Mouse embryo 2 (Smart-seq2)'), GSE59892 ('Mouse embryo 3 (Smart-seq)'), GSE162044 ('Neural crest (Smart-seq2)'), GSE132188 ('Pancreas (10x)'), GSE99933 ('Peripheral glia (Smart-seq2)'), GSE122466 ('Retinal neurons (10x)'), GSE64447 ('Skeletal stem cell (C1)'), and GSE201333 ('Tabula Sapiens (Smart-seq2/10x)').

Raw FASTQ or BAM files analyzed in this work are available from the SRA with the following accessions: SRP188993 ('BM-MNC (CITE-seq)'), SRP168426 ('Retinal neurons (10x)'), SRP200419 ('Pancreas (10x)'), SRP109011 ('Peripheral glia (Smart-seq2)'), SRP239468 ('Skeletal stem cell (C1)'), SRP094420 ('HSCs and MPPs (inDrop)'), and SRP476153 ('Mouse neurogenesis (10x)' and 'Mouse mature neural cell types (10x)').

Five expression datasets covering mouse embryogenesis periods from implantation to organogenesis are accessible from GEO or ArrayExpress with the following accessions: GSE100597 ('Implantation (E3.5-E6.5)'), GSE109071 ('Implantation (E5.5-E6.5)'), E-MTAB-6967 ('Gastrulation (E6.5-E8.5)'), GSE186069 ('Organogenesis (E8.5)'), and GSE228590 ('Organogenesis (E8.75-P0)').

The publicly available oligodendroglioma and AML expression data analyzed in this work are available with GEO accession numbers GSE70630 and GSE116256, respectively.

Reference genomes and annotation files for GRCm39 (mouse) and GRCh38.p13 (human) were obtained from Ensembl release 109 (February 2023) via the archive at https://feb2023.archive.ensembl.org.

## Research involving human participants, their data, or biological material

Policy information about studies with human participants or human data. See also policy information about sex, gender (identity/presentation), and sexual orientation and race, ethnicity and racism.

| | |
|---|---|
| Reporting on sex and gender | No human data was generated for this study. |
| Reporting on race, ethnicity, or other socially relevant groupings | N/A |
| Population characteristics | N/A |
| Recruitment | N/A |
| Ethics oversight | N/A |

Note that full information on the approval of the study protocol must also be provided in the manuscript.

# Field-specific reporting

Please select the one below that is the best fit for your research. If you are not sure, read the appropriate sections before making your selection.

☒ Life sciences  ☐ Behavioural & social sciences  ☐ Ecological, evolutionary & environmental sciences

For a reference copy of the document with all sections, see nature.com/documents/nr-reporting-summary-flat.pdf

# Life sciences study design

All studies must disclose on these points even when the disclosure is negative.

| | |
|---|---|
| Sample size | For both single-cell RNA-seq studies and mouse experiments, sample sizes were based on prior studies with similar designs and optimized for feasibility. The selected sizes were sufficient to detect consistent and biologically meaningful trends across replicates and conditions, and to support the statistical analyses presented. Where applicable, findings were validated in independent cohorts or with orthogonal methods to ensure robustness. All results were analyzed and interpreted using statistically appropriate techniques as described in Methods. |

| | |
|---|---|
| Data exclusions | Quality control metrics for data exclusion are fully described in Methods. Key exclusions included scRNA-seq samples of tumors which were derived from cell lines or for which fewer than 10 malignant cells were identified, and from these samples, non-malignant cells annotated by the author as "undifferentiated". In generating the potency atlas presented in this study, phenotypes in Tabula Sapiens with fewer than five cells for a given tissue/platform pair were excluded. |
| Replication | The CytoTRACE 2 model was developed over a portion of the curated gold standard potency atlas, then tested over fully held-out data from the remainder as well as Tabula Sapiens data not included in either cohort. To ensure replicability and generalizability, CytoTRACE 2 was also tested in a leave-clade-out framework as described in Methods. CytoTRACE 2 performance was strongly concordant across these experiments and cohorts.<br><br>All experiments were replicated three times independently. |
| Randomization | To ensure generalizability and limit any bias from the primary training cohort selection, we repeated the training and testing process of CytoTRACE 2 across three additional train/test splits, generated randomly, as detailed in Methods. Performance was strongly concordant across these experiments and cohorts.<br><br>The robustness experiments in Extended Data Figure 3 were conducted with randomization and replicated five times as described in Methods. Averages across replicates were presented with confidence intervals.<br><br>For experiments with mice, randomization was not applicable as there was no treatment involved. |
| Blinding | The investigators were not blinded to group allocation, but the training and test cohorts analyzed in this work were generated without prior knowledge of CytoTRACE 2 potency predictions. The randomization framework detailed above serves as an additional control. |

# Reporting for specific materials, systems and methods

We require information from authors about some types of materials, experimental systems and methods used in many studies. Here, indicate whether each material, system or method listed is relevant to your study. If you are not sure if a list item applies to your research, read the appropriate section before selecting a response.

### Materials & experimental systems

| n/a | Involved in the study |
|---|---|
| ☐ | ☒ Antibodies |
| ☒ | ☐ Eukaryotic cell lines |
| ☒ | ☐ Palaeontology and archaeology |
| ☐ | ☒ Animals and other organisms |
| ☒ | ☐ Clinical data |
| ☒ | ☐ Dual use research of concern |
| ☒ | ☐ Plants |

### Methods

| n/a | Involved in the study |
|---|---|
| ☒ | ☐ ChIP-seq |
| ☐ | ☒ Flow cytometry |
| ☒ | ☐ MRI-based neuroimaging |

## Antibodies

| | |
|---|---|
| Antibodies used | Immunostaining antibody:<br>anti-E-Cadherin-Alexa Fluor 488 antibody (BD Biosciences 560061, 1:50)<br><br>Flow cytometry antibodies:<br>anti-mouse lineage cocktail-A700 (BioLegend 133313, 5 μl per mouse)<br>anti-CD117 (c-Kit)-BV395 (Thermo Fisher Scientific 363-1171-80, 1:100)<br>anti-Sca1-BV605 (BioLegend 108133, 1:100)<br>anti-CD34-eFluor 450 (Thermo Fisher Scientific 48-0341-80, 1:40),<br>anti-CD16/32-BV711 (BD Biosciences 740659, 1:100)<br>anti-CD135-BV421 (BioLegend 135313, 1:100)<br>anti-CD127 (IL-7Rα)-BV711 (BioLegend 135035, 1:100)<br>anti-CD3-BV711 (BioLegend 100241, 1:100)<br>anti-CD8a-BV605 (BioLegend 100743, 1:100)<br>anti-CD19-BV605 (BioLegend 115539, 1:100) |
| Validation | All antibodies used were validated by the respective manufactures. The validation statement of the antibodies on the manufacture's website can be found below.<br>anti-E-Cadherin-Alexa Fluor 488 antibody (https://www.bdbiosciences.com/en-us/products/reagents/microscopy-imaging-reagents/immunofluorescence-reagents/alexa-fluor-488-mouse-anti-e-cadherin.560061?tab=product_details), anti-mouse lineage cocktail-A700 (https://www.biolegend.com/en-us/products/alexa-fluor-700-anti-mouse-lineage-cocktail-with-isotype-ctrl-8122), anti-CD117 (c-Kit)-BV395 (https://www.thermofisher.com/antibody/product/CD117-c-Kit-Antibody-clone-2B8-Monoclonal/363-1171-80), anti-Sca1-BV605 (https://www.biolegend.com/en-us/products/brilliant-violet-605-anti-mouse-ly-6a-e-sca-1-antibody-8664), anti-CD34-eFluor 450 (https://www.thermofisher.com/antibody/product/CD34-Antibody-clone-RAM34-Monoclonal/48-0341-80), anti-CD16/32-BV711 (https://www.bdbiosciences.com/en-us/products/reagents/flow-cytometry-reagents/research-reagents/single-color-antibodies-ruo/bv711-rat-anti-mouse-cd16-cd32.740659?tab=product_details), anti-CD135-BV421 (https://www.biolegend.com/en-us/products/brilliant-violet-421-anti-mouse-cd135-antibody-8728), anti-CD127 (IL-7Rα)-BV711 (https:// |

www.biolegend.com/en-us/products/brilliant-violet-711-anti-mouse-cd127-il-7ralpha-antibody-10632), anti-CD3-BV711 (https://www.biolegend.com/en-us/products/brilliant-violet-711-anti-mouse-cd3-antibody-10022), anti-CD8a-BV605 (https://www.biolegend.com/en-us/products/brilliant-violet-605-anti-mouse-cd8a-antibody-7636), and anti-CD19-BV605 (https://www.biolegend.com/en-us/products/brilliant-violet-605-anti-mouse-cd19-antibody-7645)

# Animals and other research organisms

Policy information about studies involving animals; ARRIVE guidelines recommended for reporting animal research, and Sex and Gender in Research

| | |
|---|---|
| Laboratory animals | 8- to 12-week-old C57BL/6 mice were used. |
| Wild animals | The study did not involve wild animals. |
| Reporting on sex | Equal numbers of males and females were used. |
| Field-collected samples | This study did not involve samples collected from the field. |
| Ethics oversight | All animal procedures were conducted according to a protocol approved by the Stanford University APLAC committee (10868). |

Note that full information on the approval of the study protocol must also be provided in the manuscript.

# Plants

| | |
|---|---|
| Seed stocks | *Report on the source of all seed stocks or other plant material used. If applicable, state the seed stock centre and catalogue number. If plant specimens were collected from the field, describe the collection location, date and sampling procedures.* |
| Novel plant genotypes | *Describe the methods by which all novel plant genotypes were produced. This includes those generated by transgenic approaches, gene editing, chemical/radiation-based mutagenesis and hybridization. For transgenic lines, describe the transformation method, the number of independent lines analyzed and the generation upon which experiments were performed. For gene-edited lines, describe the editor used, the endogenous sequence targeted for editing, the targeting guide RNA sequence (if applicable) and how the editor was applied.* |
| Authentication | *Describe any authentication procedures for each seed stock used or novel genotype generated. Describe any experiments used to assess the effect of a mutation and, where applicable, how potential secondary effects (e.g. second site T-DNA insertions, mosiacism, off-target gene editing) were examined.* |

# Flow Cytometry

## Plots

Confirm that:

☒ The axis labels state the marker and fluorochrome used (e.g. CD4-FITC).

☒ The axis scales are clearly visible. Include numbers along axes only for bottom left plot of group (a 'group' is an analysis of identical markers).

☒ All plots are contour plots with outliers or pseudocolor plots.

☒ A numerical value for number of cells or percentage (with statistics) is provided.

## Methodology

| | |
|---|---|
| Sample preparation | Hips, femurs, tibia, and humeri were harvested from C57BL/6 mice. Bones were cleaned, cut, and flushed with a syringe filled with ice-cold FACS buffer (2% fetal bovine serum [FBS] in Hanks' Balanced Salt Solution [HBSS] buffer). Cells in FACS buffer were filtered through a 40 μm filter, pelleted, and then incubated in ammonium-chloride-potassium (ACK) lysis buffer for 5 minutes on ice. Cells were then spun down and resuspended in 400 μl FACS buffer per mouse. Lineage depletion beads (Miltenyi Biotec 130-110-470) were added to the cells (50 μl per mouse) and incubated for 10 min at 4°C. After incubation, the cells were loaded onto an LS magnetic separation column (Miltenyi Biotec 130-042-401), which was subsequently washed with 3 × 3 mL of FACS buffer. Before and after washing, pass-through cells were collected, spun down, and resuspended in FACS buffer.

Blood samples were collected from the same mice for the isolation of CD8a+ T cells (CD3+ CD8a+) and B (CD19+) cells. Peripheral blood mononuclear cell (PBMC) isolation was performed using a SepMate™-15 tube (STEMCELL technologies 85415) according to the manufacturer's instructions. |
| Instrument | The cells were analyzed on a BD FACSAria II sorter. |
| Software | Data were analyzed with FlowJo V10. |

| | |
|---|---|
| Cell population abundance | The post-sort samples were re-analyzed using FACS to verify a purity level of over 95%. |
| Gating strategy | The major cell populations were first identified within the FSC/SSC plots, followed by doublet exclusion. The mouse HSC, MPP, T cell, and B cell populations were then gated according to previously published gating strategies (PMID: 33236985). Fluorescence-minus-one (FMO) controls were used to discriminate between positive and negative staining. |

☒ Tick this box to confirm that a figure exemplifying the gating strategy is provided in the Supplementary Information.

