## [Peer Review File · Nature Methods]

Improved reconstruction of single-cell developmental potential with CytoTRACE 2

Corresponding Author: Dr Aaron Newman

Version 0:

Decision Letter:

Our ref: NMETH-BC60264

7th May 2025

Dear Aaron,

Thank you for submitting your revised manuscript "Improved reconstruction of single-cell developmental potential with CytoTRACE 2" (NMETH-BC60264). It has now been seen by the original referees and their comments are below and after considering the reviewer reports, we'll be happy in principle to publish it in Nature Methods, pending minor revisions to satisfy the referees' final requests and to comply with our editorial and formatting guidelines. :)

TRANSPARENT PEER REVIEW

ORCID

Sincerely,
Madhura

Madhura Mukhopadhyay, PhD

Reviewer #1 (Remarks to the Author):

CytoTRACE2 addresses two key problems exist in the field currently with existing methods to measure cellular potency or stemness – variation in the number of genes profiled per cell due to differences in single-cell sequencing platform used (such as when 10X and SMART-seq data is combined) and the lack of an absolute measure of potency (which the authors refer to as “absolute ordering”) that allows cells across two datasets to be compared. Especially in the case of the latter, batch-correction based approaches may not suffice to render potency measures comparable across datasets. The key innovation in CytoTRACE2 appears to be its architecture that is able to account for batch and platform-specific differences that may arise when it is run on a dataset by combined both TPM expression levels of a gene along with its ranked expression. These abilities/advantages of CytoTRACE2 are tested rigorously (via down-sampling of counts, estimation of test dataset accuracy across datasets from different platforms, etc.).

Unlike most multi-layered neural network representations, their architecture allows for interpretability that helps explain what the model is learning. This is evident in Figure 2c, where a coarse visualization of potency-associated genes suggests that the CytoTRACE2 network is, to a good degree, learning genes that are differentially expressed at different potency levels. While in principle, a differential expression (DE) analysis of data across multiple potency stages may have yielded the same potency-associated genes (assuming that the complexity of batch and platform-specific effects can be modelled using existing DE methods), combining DE genes into a potency classifier would still be a non-trivial affair. The fact that its ability to predict stemness/potency across tissue types is remarkable, esp. given that despite the bias of the training data towards epithelial and mesenchymal tissues, it is able to perform reasonably well in predicting potency amongst HSC and immune populations.

The software itself is easy to run, and we were able to confirm the accuracy of its outputs with in-house and unpublished data that we possess from mouse fetuses. We noted that CytoTRACE2 scores had some measure of co-variation with the cell cycle status of a cell, with cells with higher cell cycle gene expression on average having a higher CytoTRACE2 score. This lines up with the authors’ observation in Extended Figure 9e where mitotic spindle genes are enriched amongst potency-associated genes of totipotency (and to some extent also pluripotency and multipotency features). To an extent, this is to be expected given that (on average) embryonic and developmental cells are more proliferative early in development as opposed to late, and the labelling scheme followed during training assumes higher potency early in development as opposed to late in development or adulthood.

A key test of whether the way potency defined in this manner during training (i.e., where an adult-derived HSC is treated as totipotent and later-stage embryo-derived cells are also treated as totipotent) is a valid definition is if CytoTRACE2 can distinguish potency levels within embryonic or adult datasets. This again is demonstrated rigorously in many ways when evaluated with HSC CITE-seq data and CRISPR KO data.

In summary, CytoTRACE2 represents a major leap over CytoTRACE1 and other existing methods to measure stemness, as seen in Extended Data Figure 5c and in analyses relating to the ability to absolutely order potency levels across datasets. It sets the state-of-the-art in the field for measuring stemness/potency. The gene set binary network and UCell/AddModuleScore-based framework used in potency labelling forms a useful prototype for the field when it comes to carrying out similar labelling tasks in single-cell RNA-seq analyses. The only potential limitation is that the method lacks direct applicability to Visium, given the nature of Visium spatial data where multiple cells are represented in a single spot, but may be applicable in Xenium Prime datasets. However, this is a very minor concern that does not take away from the other contributions of this work.

Reviewer #1 (Remarks on code availability):

We have however run the code on n independent dataset

Reviewer #2 (Remarks to the Author):

Authors present a new version of CytoTRACE, a well-used toolkit that predicts differentiation states in scRNAseq data. In this version, CytoTRACE2, authors claim to make improvements in overall interpretability and developmental potential predictions. As part of this work, authors contributed a diverse and well-curated cell potency atlas of human and mouse scRNAseq data, and novel ordering metrics that contextualize cell developmental states across datasets as well as within datasets. Overall, the brief communication manuscript is compact but well written, and the updates to the original CytoTRACE should be well adapted by current CytoTRACE users. The benchmarking approaches taken are also appropriate. One aspect of the work that doesn't come through is how batch effects (whether site or seq. platform) are handled both by the method itself and in the atlas. It is noted that no explicit batch corrections or integrations were performed but without explanation. Clarifications in the text would also be appreciated in sections where human, mouse, or both

species are queried in the potency atlas (particularly lines 192-211).

Reviewer #2 (Remarks on code availability):

Code is publically available and I was able to install the r version of cytotrace2.

We thank the reviewers for their thoughtful feedback. Below we provide point-by-point responses. Reviewer comments are in *italic*, and our responses follow in regular text.

Reviewer #1:

Remarks to the Author:

CytoTRACE2 addresses two key problems exist in the field currently with existing methods to measure cellular potency or stemness – variation in the number of genes profiled per cell due to differences in single-cell sequencing platform used (such as when 10X and SMART-seq data is combined) and the lack of an absolute measure of potency (which the authors refer to as “absolute ordering”) that allows cells across two datasets to be compared. Especially in the case of the latter, batch-correction based approaches may not suffice to render potency measures comparable across datasets. The key innovation in CytoTRACE2 appears to be its architecture that is able to account for batch and platform-specific differences that may arise when it is run on a dataset by combined both TPM expression levels of a gene along with its ranked expression. These abilities/advantages of CytoTRACE2 are tested rigorously (via down-sampling of counts, estimation of test dataset accuracy across datasets from different platforms, etc.).

Unlike most multi-layered neural network representations, their architecture allows for interpretability that helps explain what the model is learning. This is evident in Figure 2c, where a coarse visualization of potency-associated genes suggests that the CytoTRACE2 network is, to a good degree, learning genes that are differentially expressed at different potency levels. While in principle, a differential expression (DE) analysis of data across multiple potency stages may have yielded the same potency-associated genes (assuming that the complexity of batch and platform-specific effects can be modelled using existing DE methods), combining DE genes into a potency classifier would still be a non-trivial affair. The fact that its ability to predict stemness/potency across tissue types is remarkable, esp. given that despite the bias of the training data towards epithelial and mesenchymal tissues, it is able to perform reasonably well in predicting potency amongst HSC and immune populations.

The software itself is easy to run, and we were to able to confirm the accuracy of its outputs with in-house and unpublished data that we possess from mouse fetuses. We noted that CytoTRACE2 scores had some measure of co-variation with the cell cycle status of a cell, with cells with higher cell cycle gene expression on average having a higher CytoTRACE2 score. This lines up with the authors’ observation in Extended Figure 9e where mitotic spindle genes are enriched amongst potency-associated genes of totipotency (and to some extent also pluripotency and multipotency features). To an extent, this is to be expected given that (on average) embryonic and developmental cells are more proliferative early in development as opposed to late, and the labelling scheme followed during training assumes higher potency early in development as opposed to late in development or adulthood.

A key test of whether the way potency defined in this manner during training (i.e., where an adult-derived HSC is treated as totipotent and later-stage embryo-derived cells are also treated as totipotent) is a valid definition is if CytoTRACE2 can distinguish potency levels within embryonic or adult datasets. This again is demonstrated rigorously in many ways when evaluated with HSC CITE-seq data and CRISPR KO data.

In summary, CytoTRACE2 represents a major leap over CytoTRACE1 and other existing methods to measure stemness, as seen in Extended Data Figure 5c and in analyses relating to the ability to absolutely order potency levels across datasets. It sets the state-of-the-art in the field for measuring stemness/potency. The gene set binary network and UCell/AddModuleScore-based framework used in potency labelling forms a useful prototype for the field when it comes to carrying out similar labelling tasks in single-cell RNA-seq analyses. The only potential limitation is that the method lacks direct applicability to Visium, given the nature of Visium spatial data where multiple cells are represented in a single spot, but may be applicable in Xenium Prime datasets. However, this is a very minor concern that does not take away from the other contributions of this work.

Remarks on code availability:

We have however run the code on n independent dataset

Response:

We thank the reviewer for their thorough review and positive appraisal of our work. We agree that a major strength of our approach is its ability to infer an absolute measure of potency across datasets, platforms, and tissue systems. We also agree that our framework has strong potential for application to single-cell spatial transcriptomics, where it could provide insights into spatial relationships between cells with varying developmental potential. Ongoing work in our laboratory is focused on adapting CytoTRACE 2 for these and other emerging applications. We look forward to sharing those results in future studies.

Reviewer #2:

Remarks to the Author:

Authors present a new version of CytoTRACE, a well-used toolkit that predicts differentiation states in scRNAseq data. In this version, CytoTRACE2, authors claim to make improvements in overall interpretability and developmental potential predictions. As part of this work, authors contributed a diverse and well-curated cell potency atlas of human and mouse scRNAseq data, and novel ordering metrics that contextualize cell developmental states across datasets as well as within datasets. Overall, the brief communication manuscript is compact but well written, and the updates to the original CytoTRACE should be well adapted by current CytoTRACE users. The benchmarking approaches taken are also appropriate. One aspect of the work that doesn't come through is how batch effects (whether site or seq. platform) are handled both by the method itself and in the atlas. It is noted that no explicit batch corrections or integrations were performed but without explanation. Clarifications in the text would also be appreciated in sections where human, mouse, or both species are queried in the potency atlas (particularly lines 192-211).

Remarks on code availability:

Code is publically available and I was able to install the r version of cytotrace2.

Response:

We thank the reviewer for their thoughtful comments and careful evaluation of our work. While CytoTRACE 2 incorporates a diversity of datasets spanning tissue systems and platforms, it does not perform explicit batch correction or integration. Instead, it mitigates batch effects through multiple mechanisms: (1) a dual encoding that combines normalized log-transformed inputs with a rank-based representation, (2) the use of corresponding enrichment metrics applied to the respective inputs (AMS and UCell), (3) the joint identification of positive and negative correlates of cell potency, (4) the diversity of phenotypes, datasets, and profiling platforms within the training set, which promotes generalizability and reduces susceptibility to dataset-specific biases, and (5) regularization techniques, including dropout layers within the neural network, to prevent overfitting to technical variation. Collectively, these strategies enhance robustness to differences in scale, distribution, and technical noise across datasets, enabling cross-platform, cross-tissue predictions of developmental potential.

We detail this in Methods and have revised the main text to state: “**Moreover, it [CytoTRACE 2] suppresses batch and platform-specific variation through multiple mechanisms, including competing representations of gene expression and training set diversity (Methods).**”.

Additionally, the analysis in lines 192–211 from the original submission (Fig. 2a–h) focuses on consensus multipotent genes across mouse and human tissues. We have revised the text for clarity (see below).

“To more deeply analyze multipotency **in mouse and human tissues** and explore the potential of CytoTRACE 2 for biomarker discovery, we next applied pathway enrichment analysis to genes ranked by feature importance.”